# Live-cell single-molecule tracking highlights requirements for stable Smc5/6 chromatin association in vivo

**Thomas J Etheridge[1]\*, Desiree Villahermosa[1], Eduard Campillo-Funollet[1], Alex David Herbert[1], Anja Irmisch[1†], Adam T Watson[1], Hung Q Dang[1], Mark A Osborne[2], Antony W Oliver[1], Antony M Carr[1], Johanne M Murray[1]\***

[1]Genome Damage and Stability Centre, School of Life Sciences, University of Sussex, Falmer, United Kingdom; [2]Chemistry Department, School of Life Sciences, University of Sussex, Falmer, United Kingdom

**Abstract** The essential Smc5/6 complex is required in response to replication stress and is best known for ensuring the fidelity of homologous recombination. Using single-molecule tracking in live fission yeast to investigate Smc5/6 chromatin association, we show that Smc5/6 is chromatin associated in unchallenged cells and this depends on the non-SMC protein Nse6. We define a minimum of two Nse6-dependent sub-pathways, one of which requires the BRCT-domain protein Brc1. Using defined mutants in genes encoding the core Smc5/6 complex subunits, we show that the Nse3 double-stranded DNA binding activity and the arginine fingers of the two Smc5/6 ATPase binding sites are critical for chromatin association. Interestingly, disrupting the single-stranded DNA (ssDNA) binding activity at the hinge region does not prevent chromatin association but leads to elevated levels of gross chromosomal rearrangements during replication restart. This is consistent with a downstream function for ssDNA binding in regulating homologous recombination.

**\*For correspondence:**
t.etheridge@sussex.ac.uk (TJE);
j.m.murray@sussex.ac.uk (JMM)

**Present address:** †Department of Dermatology, University Hospital Zürich, Switzerland

**Competing interests:** The authors declare that no competing interests exist.

## Introduction

The structural maintenance of chromosome (SMC) complexes – cohesin, condensin, and Smc5/6 – are critical for the correct organisation of chromosome architecture (*Uhlmann, 2016*). Whereas the functions of cohesin and condensin are increasingly well understood, the exact function of Smc5/6 complex remains relatively ambiguous. Smc5/6 is conserved across all eukaryotes and is best known for its role in the cellular response to DNA damage by ensuring the fidelity of homologous recombination repair (HRR) (*Murray and Carr, 2008*), (*Aragón, 2018*). Smc5/6 has been reported to promote replication fork stability (*Irmisch et al., 2009*; *Ampatzidou et al., 2006*) and facilitate DNA replication through natural pausing sites (*Menolfi et al., 2015*). Biochemically, the complex can regulate pro-recombinogenic helicases (*Xue et al., 2014*), (*Bonner et al., 2016*). It has also been proposed to monitor DNA topology (*Jeppsson et al., 2014*) and recently been shown to restrict viral transcription (*Bentley et al., 2018*; *Niu et al., 2017*). Hypomorphic mutants show significant defects in sister-chromatid HRR, display replication fork instability, are sensitive to a wide range of genotoxins, and accumulate unresolved recombination intermediates (*Irmisch et al., 2009*), (*De Piccoli et al., 2006*), (*Bermúdez-López et al., 2010*). Intriguingly, complete inactivation of the Smc5/6 complex in a variety of organisms leads to cell death, and this essential nature suggests that it possesses additional functions beyond HR (homologous recombination) as deletions of core HR factors are viable.

Like all SMC complexes, the core of Smc5/6 is composed of two folded proteins, Smc5 and Smc6, which form a heterodimer (*Figure 1A*). Each subunit comprises a long coiled-coil arm with a hinge region at one end and a globular ATPase head at the other end (*Uhlmann, 2016*). All three

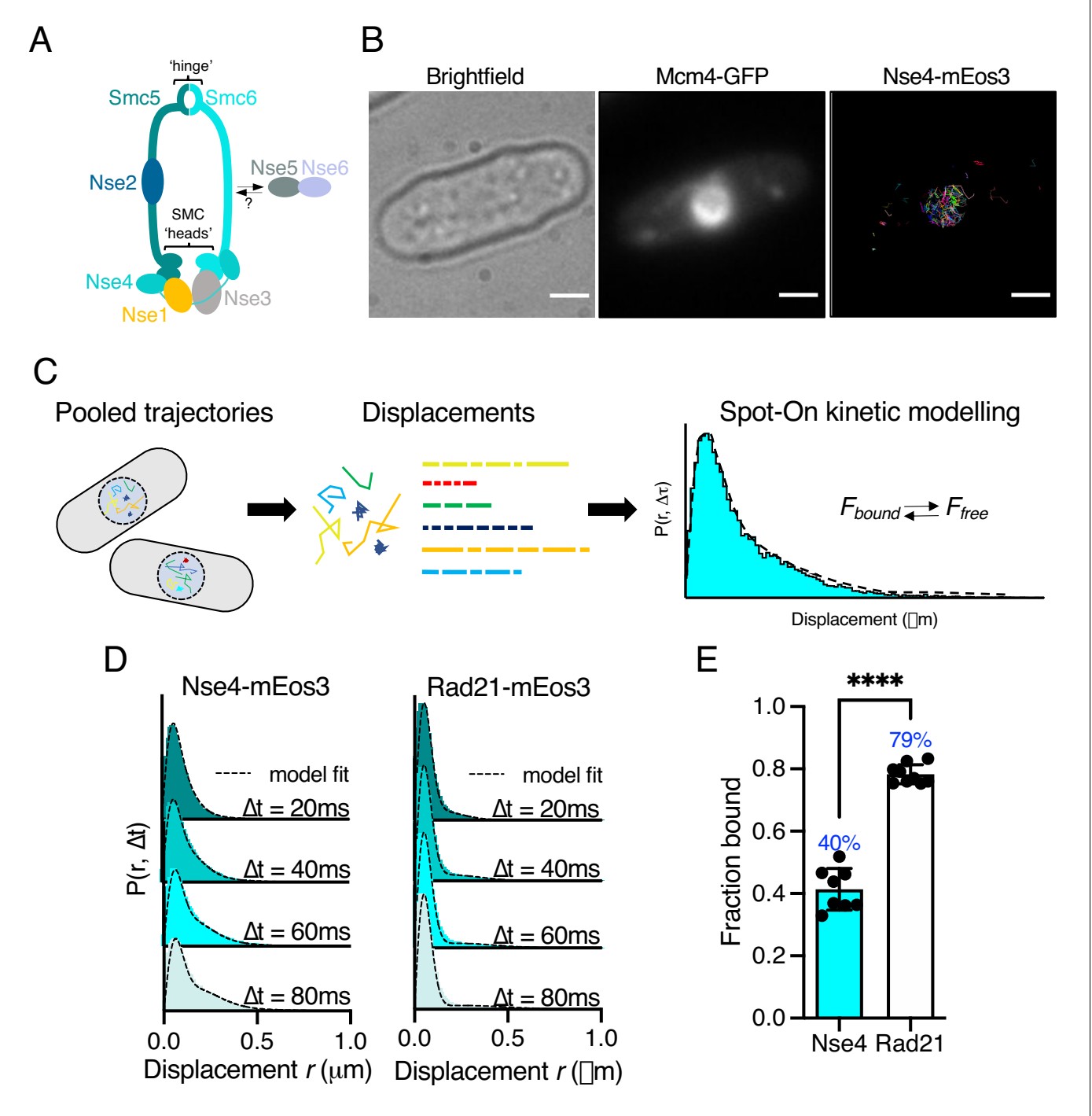

**Figure 1.** Single-particle tracking of Smc5/6 to monitor chromatin association in live cells. (**A**) Schematic representation of the Smc5/6 complex in fission yeast. (**B**) Nse4–mEos3 tracking shows nuclear localisation of trajectories. SPT trajectories demonstrated confinement within nuclear region (right) that colocalised with the nuclear replication protein Mcm4 fused to GFP. Scale bar = 2 μm. (**C**) Overview of approach to quantifying chromatin association using SPT data and Spot-On kinetic modelling. (**D**) Probability density function (PDF) histograms and Spot-On model fitting (dashed line) for Nse4–mEos3 (Smc5/6) and Rad21–mEos3 (cohesin) single-molecule displacements at different time intervals. Displacements are from three pooled independent experiments, each with three technical repeats. (**E**) Fraction-bound values derived from Spot-On model fitting. Mean (±S.D). Black dots indicate Spot-On $F_{bound}$ values derived from each technical repeat from three independent experiments. Percentages in blue denote fraction-bound value from fitting pooled data in (**D**). ****p<0.0001.

The online version of this article includes the following figure supplement(s) for figure 1:

**Figure supplement 1.** Phenotypic characterisation of mEos3-tagged SMC subunits.

*Figure 1 continued on next page*

*Figure 1 continued*

**Figure supplement 2.** Outline of the single-particle tracking technique.
**Figure supplement 3.** Smc5/6 behaviour fits a three-state model.
**Figure supplement 4.** Single-particle tracking of SMC complex subunits.

SMC heterodimers interact at the hinge and ATP binding/hydrolysis occurs in two pockets formed between the heads of the two subunits. For all SMC complexes, ATP turnover is essential for cell viability and has been proposed to bring about conformational changes in the arms (*Hirano and Hirano, 2006*), (*Diebold-Durand et al., 2017*; *Muir et al., 2020*). The ATPase activity is also key to the interaction of SMC's with DNA: cohesin's ATPase is required for both loading and dissociation from DNA (*Elbatsh et al., 2016*), whilst condensin is dependent on its ATPase activity for translocating along DNA and forming loop structures (*Terakawa et al., 2017*), (*Elbatsh et al., 2019*). The role of the Smc5/6 ATPase in DNA association has not been studied in detail.

The Smc5/6 hinge contains specialised interfaces that are important for interacting with single-stranded DNA (ssDNA) (*Alt et al., 2017*). Disruption of these regions by mutation results in sensitivity to DNA damaging agents. The Smc5/6 ATPase heads are bridged by a sub-complex of three non-SMC elements (Nse), Nse4 (kleisin), and two kleisin-interacting tandem winged-helix element (KITE) proteins, Nse1 and Nse3. Nse1 has a RING finger and, in association with Nse3, has been shown to have ubiquitin ligase activity (*Doyle et al., 2010*). The winged-helix domain of Nse3 possesses double-stranded DNA (dsDNA) binding activity, which is essential for viability (*Zabrady et al., 2016*). The dsDNA binding has been predicted to provide the basis for initial chromatin association and loading of the complex (*Zabrady et al., 2016*). In addition to the Nse1/3/4 subcomplex, Nse2, a SUMO ligase, is associated with the Smc5 coiled-coil arm. DNA association of the Smc5/6 complex is required to activate the Nse2 SUMO ligase, which SUMOylates a range of targets within and outside of the complex (*Varejão et al., 2018*). Two further proteins, Nse5 and Nse6, also associate with the Smc5/6 complex in yeasts (both *Saccharomyces cerevisiae* and *Schizosaccharomyces pombe*). However, unlike the other Nse proteins, Nse5 and Nse6 have not been identified as part of a Smc5/6 holo-complex in human cells (*Pebernard et al., 2006*), (*Taylor et al., 2008*).

Chromatin loading of the structurally related cohesin complex requires accessory proteins, the cohesin-loader complex Scc2–Scc4 (*sp*Mis4–Ssl3) (*Ocampo-Hafalla and Uhlmann, 2011*). A loading complex for Smc5/6 has not yet been defined, but recent work in fission yeast has shown that its recruitment to sites of replication fork collapse occurs via a multi-BRCT domain protein, Brc1 (*Oravcová et al., 2018*). Brc1 binds to γ-H2A and interacts with the Nse5–Nse6 subcomplex (which associates with Smc5/6 but is not part of the core complex), providing a potential mechanism by which Smc5/6 is recruited and loaded. In *S. cerevisiae*, the N-terminal four BRCT domains of the Brc1 homologue, Rtt107, have also been shown to bind Nse6 amongst a number of other proteins in the DNA damage response (*Wan et al., 2019*). In human cells, recruitment of Smc5/6 to interstrand cross-links was shown to depend on interactions between SLF1 – another multi-BRCT domain protein – and SLF2 – a distant homologue of Nse6 (*Räschle et al., 2015*). These observations suggest that recruitment of Smc5/6 through Nse6 and a BRCT-domain mediator protein has been conserved through evolution.

Understanding how Smc5/6 is recruited to, and associates with, the chromatin is an important step in defining how it regulates recombination processes and other potential DNA transactions. To date, the study of Smc5/6 chromatin association has been mostly limited to chromatin immunoprecipitation (ChIP)-based methodologies. Recent studies have shown single-particle tracking (SPT) microscopy can provide robust measurements of chromatin interacting proteins in vivo and offer complementary data to genome-wide approaches.

Here, we perform SPT using photoactivated localisation microscopy (PALM) in live fission yeast cells to monitor chromatin association of Smc5/6. Using a range of *smc* and *nse* mutants, we investigated the role of its ATPase activity, DNA interaction sites, and protein binding partners in promoting chromatin association. This highlighted that ATPase activity and dsDNA binding are both crucial for chromatin association. In contrast, interaction with ssDNA at the hinge is not required for stable chromatin loading, but we show that it is important to prevent gross chromosomal rearrangements at collapsed replication forks. We also establish that the Nse5–Nse6 sub-complex is required for

almost all chromatin association, whereas Brc1 is required for only a proportion of the association. These data define the Brc1–Nse6-dependent sub-pathway of chromatin interaction and identify parallel Nse6-dependent but Brc1-independent sub-pathway(s).

## Results

### Smc5/6 is chromatin associated in unchallenged cells

To monitor Smc5/6 chromatin association in living yeast cells we used photoactivated localisation microscopy combined with SPT (*Manley et al., 2008*). We created a fission yeast strain that endogenously expressed the kleisin subunit Nse4 fused to the photoconvertible fluorophore mEos3 and verified this allele had no measurable impact on cellular proliferation (*Figure 1—figure supplement 1*). We imaged photoconverted subsets of Nse4–mEos3 in live yeast cells at high temporal resolution (20 ms exposure) and created trajectories by localising and tracking individual fluorophores (*Figure 1—figure supplement 2*). Nse4–mEos3 localisations and trajectories showed nuclear confinement consistent with previous studies (*Pebernard et al., 2008*; *Figure 1B*).

To evaluate the chromatin association of Smc5/6 from our data, we used the recently described 'Spot-On' software (*Hansen et al., 2018*) (see Materials and methods). Spot-On implements a bias-aware kinetic modelling framework and robustly extracts diffusion constants and sub-populations from histograms of the molecular displacements that make up each trajectory (*Figure 1C*). We tracked Nse4–mEos3 in asynchronous live cells and created displacement histograms over four time intervals (*Figure 1D*). The profiles show a clear peak of short displacements (<100 nm) indicative of a chromatin-bound fraction of Nse4–mEos3 in unchallenged cells. Spot-On kinetic modelling revealed a fraction bound of about 40% (*Figure 1E*). The displacement distributions were best described with a three-state fit which, in addition to bound and freely diffusing species, included an intermediate slow-diffusing population. This may describe transient interactions with chromatin or anomalous diffusion as a result of a crowded molecular environment (*Woringer et al., 2020*; *Figure 1—figure supplement 3*, Materials and methods). Tracking of other core Smc5/6 (Nse2 and Smc6) subunits revealed similar displacement profiles and bound fractions, suggesting the dynamics of the kleisin subunit is indicative of the whole complex (*Figure 1—figure supplement 4*).

We next compared Smc5/6 chromatin association with the structurally related cohesin complex. As fission yeast cells reside in G2 for the majority of the cell cycle, we hypothesised that cohesin would be stably associated with the chromatin (*Bernard et al., 2008*) and should thus demonstrate a higher fraction bound. As predicted, tracking of Rad21 (kleisin) and Smc1 (arm) fused to mEos3 revealed displacement profiles with greater proportions of short displacements compared to Smc5/6 subunits and subsequently resulted in greater bound fractions extracted from Spot-On model fitting (*Figure 1D,E*, *Figure 1—figure supplement 4*). These observations show that interaction of cohesin and Smc5/6 with chromatin is distinct and different and suggests that their association occurs with different dynamics.

### dsDNA binding is required for efficient chromatin association

Smc5/6 has been shown to bind both ds- and ssDNA. The KITE protein Nse3 has a dsDNA binding domain in both humans and fission yeast and is situated at the head end of the complex (*Figure 2A*). This activity is essential and was predicted to be the initial point of interaction between Smc5/6 and the chromatin required before loading (*Zabrady et al., 2016*). To assess whether Nse3 dsDNA interaction plays a role in global chromatin association, we tracked Nse4–mEos3 in a *nse3-R254E* genetic background. This hypomorphic mutation has been shown to disrupt but not fully abolish dsDNA binding by Nse3 (*Zabrady et al., 2016*). When compared to *nse3+*, Nse4–mEos3 displacement histograms from asynchronous *nse4-mEos3 nse3-R254E* cells showed a broader profile suggesting the complex had become more dynamic (*Figure 2B,C*). This resulted in a reduction in the fraction-bound value in Spot-On analysis (*Figure 2D*). This confirms in vivo that dsDNA binding by Nse3 underpins the chromatin association of Smc5/6.

### Smc5/6 ATPase activity is required for efficient chromatin association

Each of the SMC complexes possess ATPase activity, with two separate and distinct active sites within juxtaposed 'head' domains, which are generated by bringing together the required signature

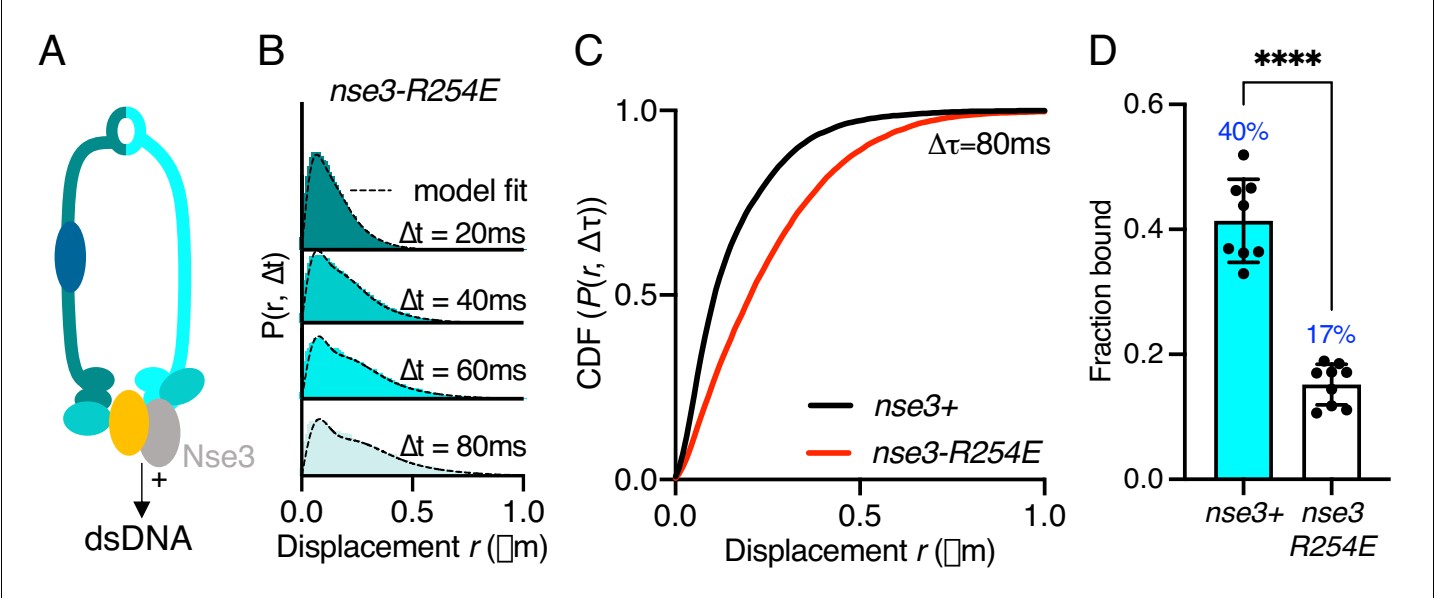

**Figure 2.** Stable Smc5/6 chromatin association requires dsDNA binding activity. (A) Schematic representation of the region of known dsDNA interaction in *S. pombe* Smc5/6. (B) Probability density function histogram of pooled Nse4–mEos3 single-molecule in *nse3-R254E* background and Spot-On model fitting (dashed line). The resulting fraction of bound molecules compared to wild-type data in *Figure 1D*. Bar chart shows mean ± S.E. M. Black dots denote independent repeats. ***p=0.0003. (C) Cumulative distribution function (CDF) of pooled Δt = 80 ms data from (B). (D) $F_{bound}$ values derived from Spot-On model fitting of Nse4–mEos3 in *nse3-R254E* background. Black dots denote each technical repeat from three independent experiments. Percentages in blue denote fraction-bound value from fitting pooled data in (B). Mean (±S.D). ****p<0.0001.

motifs in trans (*Figure 3A*). Like all SMC complexes the ATPase activity of Smc5/6 is essential and inactivating mutations in either of the two Walker motifs are non-viable (*Verkade et al., 1999*; *Fousteri and Lehmann, 2000*). Therefore, to investigate the influence of ATPase activity on chromatin association of the Smc5/6 complex, we first mutated the 'arginine finger' of Smc5 (*smc5-R77A*) or Smc6 (*smc6-R150A*). Mutation of the equivalent residues in other SMC complexes does not typically affect the basal level of ATP turnover, but instead acts to abolish stimulation of activity by DNA interaction (*Lammens et al., 2004*). Both the *smc5-R77A* and the *smc6-R150A* mutation resulted in sensitivity to replication stress (*Figure 3—figure supplement 1*). Tracking of Nse4–mEos3 in these genetic backgrounds revealed increased single-molecule displacements and subsequent decreases in chromatin association of the Smc5/6 complex (*Figure 3B,C* and *Figure 3—figure supplement 1*). *smc6-R150A* led to a dramatic decrease in chromatin association, whereas mutation of the Smc5 arginine was noticeably less detrimental. Interestingly, the reduction in the levels of chromatin association correlated with sensitivity to exogenous genotoxic agents, strongly suggesting that DNA-dependent ATP hydrolysis by the two binding pockets is not equivalent.

The Smc6 arginine finger mutant was of particular interest to us as the well characterised *smc6-74* allele maps to the next residue, A151T (*Irmisch et al., 2009*), (*Verkade et al., 1999*), (*Verkade et al., 1999*), (*Outwin et al., 2009*). Single-particle tracking showed this mutant to have a similar decrease in chromatin association to *smc6-R150A*. Sequence-threaded homology models for the head domain of *S. pombe* Smc6 and comparison to the X-ray crystal structure of the head domain from *Pyrococcus furiosus* SMC in complex with ATP (PfSMC, PDB: 1XEX) allowed us to create specific mutations designed to display a graduated effect on the Smc6 arginine finger: Thr135 in Smc6 was mutated to a series of hydrophobic amino acids with increasing size, each predicted to produce increasingly severe steric clashes with the arginine finger when engaged in interaction with bound ATP (*Figure 3D*).

Phenotypic analysis of each *smc6* mutant confirmed that the predicted severity of steric clash (Phe > Leu > Val) closely correlated with an increase in sensitivity to a range of genotoxic agents (*Figure 3E*), culminating with the most severe mutation, T135F, producing a phenotype similar to the well characterised *smc6-74* (A151T) mutant. SPT data revealed that increasing the severity of the

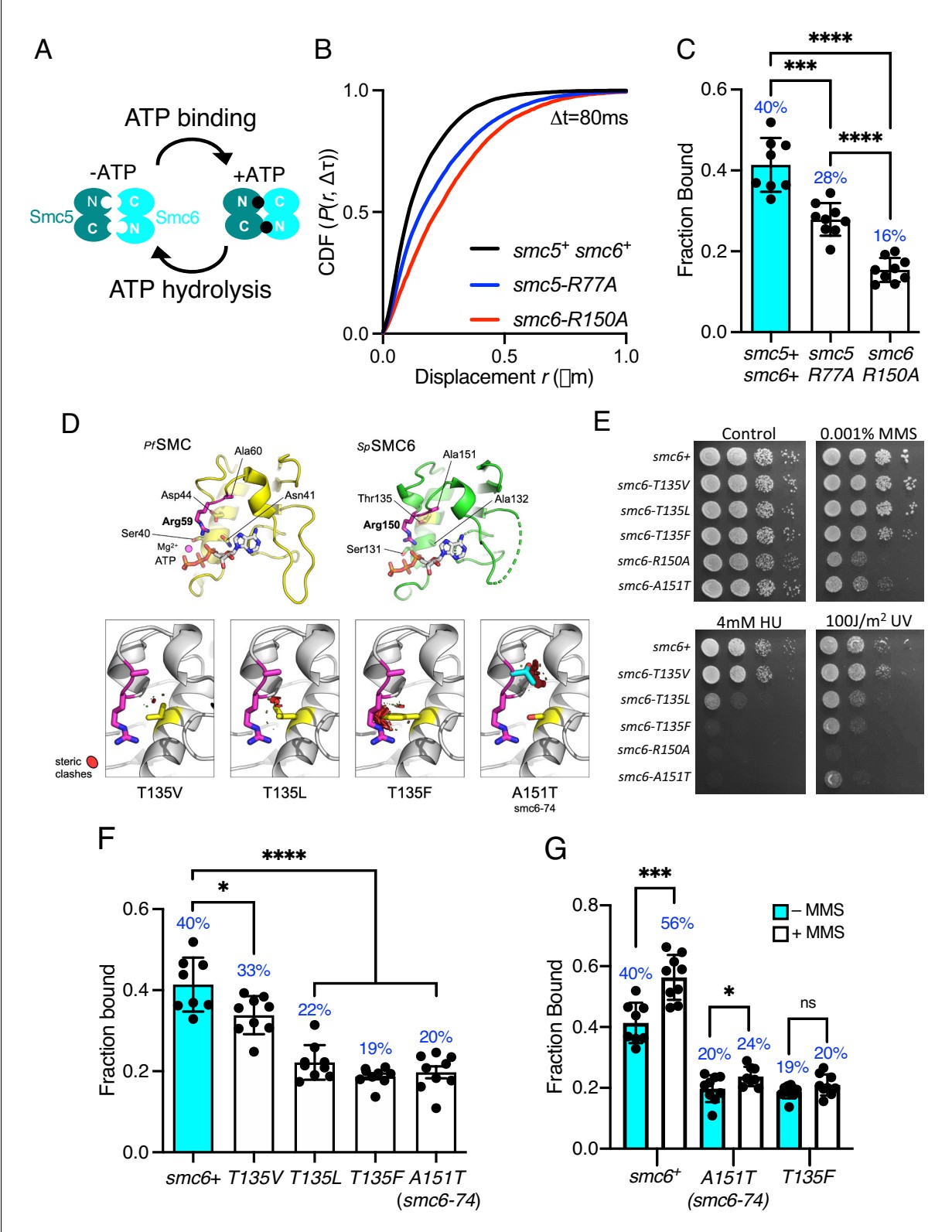

**Figure 3.** Smc5/6 ATPase activity regulates chromatin association. (**A**) Schematic representation of SMC head engagement upon ATP binding. (**B**) CDF of pooled Δt = 80 ms single-molecule displacements of Nse4–mEos3 in *smc5+ smc6+*, *smc6-R150A*, and *smc5-R77A* genetic backgrounds. (**C**) Comparison of the fraction of bound molecules from Nse4–mEos3 sptPALM experiments in asynchronous *smc6-R150A* and *smc5-R77A* genetic backgrounds to wild=type data from *Figure 1D*. Black dots denote each technical repeat from three independent experiments. Percentages in blue

*Figure 3 continued on next page*

*Figure 3 continued*

denote fraction-bound value from fitting pooled data. Mean (±S.D). ***p=0.0001, ****p<0.0001. (D) Secondary structure molecular cartoons of homology models for the head domains of *S. pombe* Smc6, highlighting the arginine finger and its interaction with ATP. The X-ray crystal structure for the head domain of *Pyrococcus furiosus SMC* in complex with ATP served as a reference, providing the expected position of bound ATP the homology model. Key amino acids are shown in 'stick representation'. The lower panel shows the predicted increase in severity of steric clashes made with the arginine finger through introduction of each of the indicated mutations. (E) Yeast spot assay of *S. pombe* strains harbouring different *smc6* ATPase mutations grown at 30°C for 3 days. (F) Fraction-bound values in each of the *smc6-T135* mutant backgrounds compared to wild-type data from *Figure 1D* and *smc6-74* (A151T). Black dots denote each technical repeat from three independent experiments. Percentages in blue denote fraction-bound value from fitting pooled data. Mean (±S.D). *p=0.0158, ****p<0.0001. (G) Fraction-bound values derived from SPT analysis of MMS-treated (0.03%, 5 hr) cells compared to asynchronous untreated data in (F). *p=0.0495, ***p=0.0005.

The online version of this article includes the following figure supplement(s) for figure 3:

**Figure supplement 1.** Characterisation of mEos3-tagged Smc5/6 ATPase mutants.

**Figure supplement 2.** Spot-On analysis of Nse4 chromatin association during 5 hr methyl methanesulfonate (MMS) treatment.

substitution corresponded with a decrease in the fraction of bound Smc5/6 (*Figure 3F*, *Figure 3—figure supplement 1*). The *smc6-T135F* strain showed similar levels of bound complex as the *smc6-74* mutation.

Since mutations in the ATPase domains render cells sensitive to replication stress (*Figure 3E*), we monitored whether these mutants could recruit the complex to chromatin after treatment with methyl methanesulfonate (MMS). Acute exposure to 0.03% MMS for 5 hr resulted in a modest increase in the fraction of Nse4–mEos3 bound to the chromatin in cells with a wild-type background (*Figure 3G*, *Figure 3—figure supplement 2*). However, in contrast both the *smc6-74* and *smc6-T135F* alleles significantly reduced or prevented Smc5/6 from being recruited to chromatin in response to MMS.

Together these data demonstrate that the ability to stimulate Smc5/6 ATPase activity through the arginine finger is crucial for its stable association with the chromatin. The disparity in phenotype between *smc6* and *smc5* ATPase mutants suggests that there could be an underlying asymmetry in the use for the two ATP binding sites, a phenomenon that has been recently described for both condensin and cohesin (*Elbatsh et al., 2016*; *Hassler et al., 2019*).

## ssDNA binding is dispensable for Smc5/6 chromatin association

We recently determined the structure of the *S. pombe* Smc5/6 hinge and demonstrated its preferential binding to ssDNA (*Alt et al., 2017*). Specialised features known as the 'latch' and 'hub' are required for efficient association with ssDNA (*Figure 4A*). The kinetics of this interaction are biphasic and appear to involve two distinct interaction points. Like mutants compromised for dsDNA binding, mutations in these key regions that weaken the interaction with ssDNA render cells viable but sensitive to replication stress and DNA damaging agents (*Alt et al., 2017*). We tested whether the ability to interact with ssDNA affected the ability of Smc5/6 to associate with chromatin.

Previously characterised mutations were introduced into the Nse4–mEos3 strain that affect either initial ssDNA interaction (*smc5-R609E R615E*), stable hinge heterodimerisation (*smc5-Y612G*), or secondary ssDNA interactions at the Smc6 hub (*smc6-F528A*, *smc6-R706C*) (*Alt et al., 2017*; *Figure 4A*, right). Spot-On model fitting to sptPALM data showed that, unlike the dsDNA binding and ATPase mutants, disruption of ssDNA interactions did not alter the bound fraction of Smc5/6 in unchallenged cells (*Figure 4B*).

Since these mutations render cells sensitive to replication stress, we monitored recruitment of Smc5/6 complex to chromatin after treatment with MMS. Disruption of ssDNA interactions either reduced, or prevented, further Smc5/6 from being recruited to chromatin in response to MMS (*Figure 4—figure supplement 1*). Together, these data show that, while dsDNA binding is required for stable association of the Smc5/6 complex with chromatin, its interactions with ssDNA are not. This is consistent with ssDNA interactions, playing a role in processes downstream of loading and we speculate that it may be important for Smc5/6 retention on the DNA during DNA repair-associated processes.

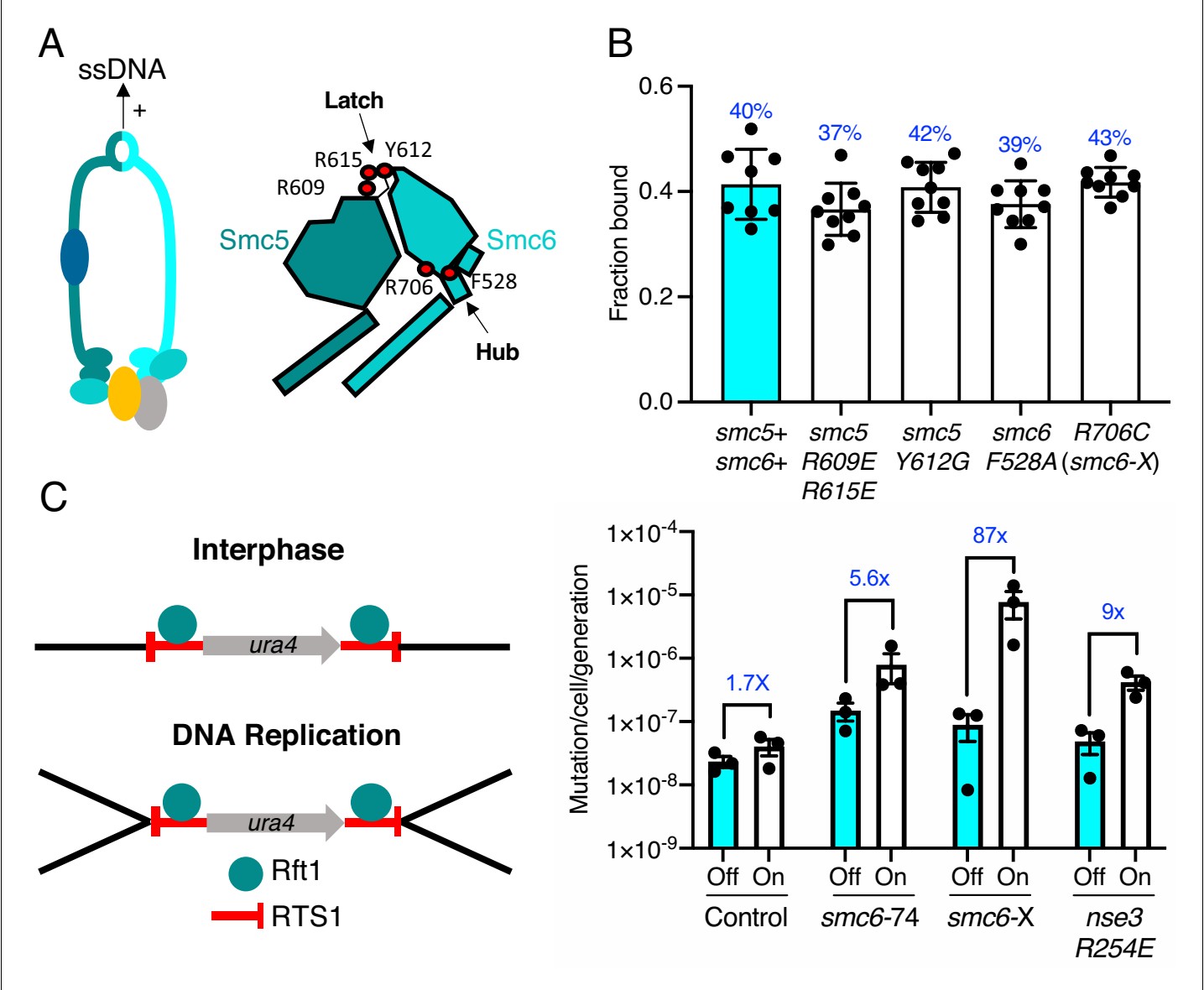

**Figure 4.** ssDNA interactions are required to prevent gross chromosomal re-arrangements but dispensable for stable Smc5/6 chromatin association. (**A**) Left: Schematic representation of the hinge region known to interact with ssDNA interaction in Smc5/6. Right: Schematic diagram of the *S. pombe* hinge region adapted from *Alt et al., 2017*. Residues implicated in ssDNA interaction are highlighted with red filled circles. (**B**) Fraction-bound values of Nse4–mEos3 derived from SPT experiments in Smc5/6 hinge mutant backgrounds compared to wild-type data from *Figure 1D*. Mean ±S.D. Black dots denote independent repeats and percentages in blue denote fraction-bound value from fitting pooled data from all repeats. (**C**) Diagram of the site-specific replication stall system *RTS1-ura4-RTS1* (*Lambert et al., 2005*), which consists of two inverted *RTS1* sequences integrated on either sides of the *ura4* gene. Rtf1 binds the *RTS1* sequence and stalls incoming replication forks coming from both centromeric and telomeric sides. Rtf1 is expressed under the control of the *nmt41* promoter which is 'off' in the presence of thiamine and 'on' upon thiamine removal. (**D**) Induction of *rtf1* in cells harbouring *RuraR* construct induces *ura4* marker loss as assayed by 5-fluoroorotic acid (5-FOA) resistance. Cells growing in the presence (Off, arrest repressed) or absence (On, arrest induced) of thiamine were analysed by fluctuation analysis. Mean ± S.E.M. Black dots denote independent repeats.

The online version of this article includes the following figure supplement(s) for figure 4:

**Figure supplement 1.** Spot-On analysis of MMS-treated ssDNA interaction mutants.

**Figure supplement 2.** Analysis of the consequences of site-specific replication fork stalling on cell viability and gross chromosomal re-arrangements.

## ssDNA interaction is required to prevent gross chromosomal rearrangements

We hypothesised that the loss of Smc5/6 chromatin association would produce distinct outcomes during HR-dependent processes compared to the loss of ssDNA interaction. To investigate this, we compared the effect of Smc5/6 mutations in the response to replication fork stalling in the previously characterised 'RuraR' replication fork barrier system (*Lambert et al., 2005*).

In fission yeast, binding of Rtf1 to the replication termination sequence, *RTS1*, arrests replication forks in a polar manner (*Dalgaard and Klar, 2001*). In the *RuraR* system, two copies of *RTS1* are placed in an inverted orientation on either side of the *ura4* marker on chromosome III (*Figure 4C*). The *RTS1* barrier activity is regulated by placing *rtf1* under the control of the *nmt41* promoter and induction of *rtf1* expression leads to arrest of replication forks converging on both *RTS1* sequences. Replication of the intervening *ura4* requires homologous recombination-dependent replication restart, which can result in genome instability via non-allelic homologous recombination (NAHR) (*Mizuno et al., 2013*) or small-scale errors by the error prone restarted fork (*Iraqui et al., 2012*). The loss of *ura4* in the *RuraR* system provides a readout that is particularly useful to characterise NAHR events. In the absence of key HR factors, such as Rad51, induction of arrest leads to viability loss, whereas mis-regulation of HR generates aberrant outcomes (*Lambert et al., 2005*).

We introduced the *smc6-R706C* (*smc6-X*) and *smc6-A151T* (*smc6-74*) mutations into the *RuraR* system. There was no loss of viability when stalling was induced at *RTS1* in these backgrounds compared to *rad51Δ* (*Figure 4—figure supplement 2*). This is consistent with Smc5/6 regulating recombination, rather than being core to the recombination process (*Murray and Carr, 2008*). Induction of replication arrest led to an increase in the loss of *ura4* activity in *smc6+*, *smc6-74*, and *smc6-X* backgrounds. There was only a modest change in the ATPase mutant (*smc6-74*) (5.6-fold) compared to *smc6+* (1.7-fold), suggesting that reduced chromatin association only moderately effects HR fork restart. To confirm this further, we introduced the *nse3-R254E* mutation into the *RuraR* strain and found similar results (ninefold) (*Figure 4C*).

Introduction of the hinge mutant (*smc6-X*) resulted in a highly elevated induction of *ura4* loss, an 87-fold increase over the uninduced (*Figure 4C* and *Supplementary file 1*). Analysis of *ura4⁻* colonies isolated after replication stalling from *smc6-X* and *smc6-74* mutants showed that most were full deletions of the intervening sequence between the two *RTS1* loci (*Figure 4—figure supplement 2*). Thus, these results highlight the ssDNA-binding region of the Smc5/6 hinge as particularly important for the suppression of NAHR and gross chromosomal rearrangements, and that stable recruitment of a defective complex (*smc6-X*) is more detrimental at collapsed replication forks than reduced Smc5/6 chromatin association (*smc6-74* and *nse3-R254E*).

## Different requirements for Nse6 and Brc1 for recruitment of Smc5/6

Recent work in fission yeast has shown that the Nse6 subunit and the BRCT-containing protein Brc1 are required for the recruitment of Smc5/6 to distinct nuclear foci in response to DNA damage (*Oravcová et al., 2018*; *Figure 5A*). To investigate whether these factors influence recruitment of the Smc5/6 complex to chromatin in unchallenged cells, the genes encoding Brc1 and Nse6 were deleted in the Nse4–mEos3 strain and Smc5/6 chromatin association monitored by SPT.

Deletion of either *brc1* or *nse6* resulted in an altered displacement profile and a concurrent decrease in the fraction of bound molecules (*Figure 5B,C*). In *brc1Δ*, the amount of chromatin-associated Smc5/6 decreased by approximately 35%, showing that only a proportion of Smc5/6 chromatin association is dependent on Brc1. Recruitment of Brc1 to chromatin is reported to be via a specific interaction with γ-H2A (*Williams et al., 2010*). We therefore investigated Smc5/6 complex recruitment in the absence of H2A phosphorylation. Introduction of *nse4-mEos3* into *hta1-SA hta2-SA* mutant cells revealed a statistically significant reduction in the fraction bound, similar to that seen in *brc1Δ* cells (*Figure 5—figure supplement 1*). These data are consistent with Brc1-dependant loading of Smc5/6 being largely confined to regions of γ-H2A.

In contrast, deletion of *nse6* showed significant deviation from the wild-type data, resulting in an almost complete loss of chromatin-associated Nse4 (*Figure 5C*), strongly supporting a Brc1-independent role for Nse6 in the stable recruitment of Smc5/6 to the chromatin. It should be emphasised that *nse6* deleted *S. pombe* cells are slow growing and very sensitive to genotoxins, whereas deletion of genes encoding proteins in the core complex is inviable. Deletion of *brc1* in an *nse6Δ*

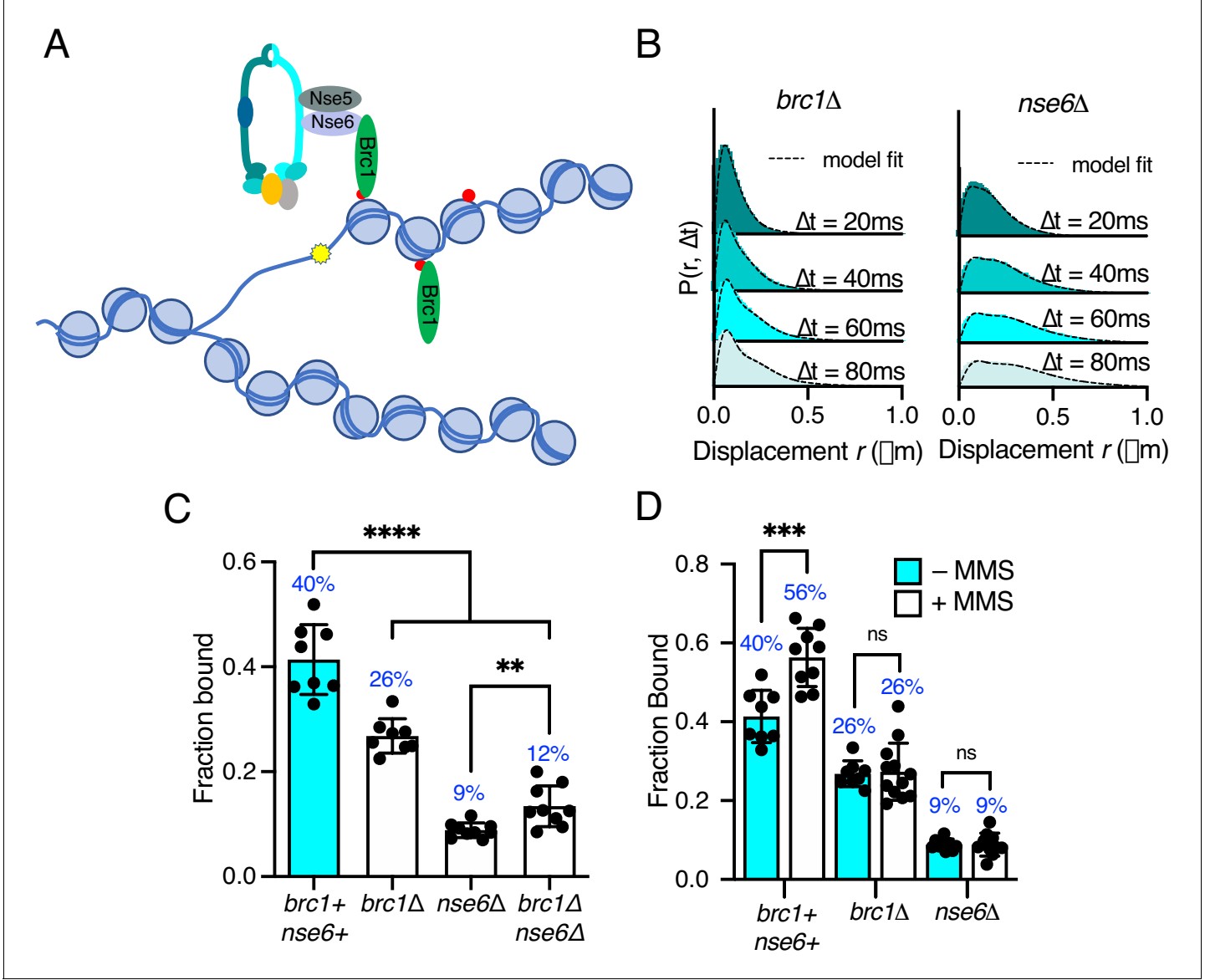

**Figure 5.** Differential requirements of Nse6 and Brc1 for Smc5/6 chromatin association. (A) Schematic diagram of Smc5/6 recruitment to γ-H2A (red dots: H2A phosphorylation) at stalled replication forks. Brc1 binds to γ-H2A and recruits Smc5/6 via an interaction with Nse6. Yellow star indicates a DNA lesion. (B) Displacement PDF histograms from asynchronous cells expressing Nse4–mEos3 in *brc1Δ* and *nse6Δ* genetic backgrounds. Data are from three pooled independent experiments, each with three technical repeats. Spot-On model fit is denoted by dashed line. (C) Comparison of Nse4–mEos3 $F_{bound}$ values derived from Spot-On fitting of SPT displacement histograms in wild type, *brc1Δ*, *nse6Δ*, and *brc1Δ nse6Δ* genetic backgrounds. Mean ± S.D. Black dot values derived from independent technical repeats; percentages in blue denote fraction-bound value from fitting pooled data from all repeats. ****p<0.0001, **p=0.0043. (D) $F_{Bound}$ fraction values from *brc1Δ and nse6Δ* cells in (C) compared to parallel experiments where cells were treated with 0.03% MMS for 5 hr. ***p<0.005, *ns* = not significant.

The online version of this article includes the following figure supplement(s) for figure 5:

**Figure supplement 1.** Spot-On analysis of Nse4 chromatin association in histone phosphorylation site and brc1γ-H2A interaction mutants.

background is viable and results in additive sensitivity to DNA damage and replication stress (*Oravcová et al., 2018*). This suggests that Smc5/6 can still associate with chromatin in the absence of Nse6, albeit at a severely reduced level. We hypothesise that the dsDNA binding activity of Nse3 is sufficient for this residual association with the chromatin. In support of this prediction, we were unable to generate the *nse6Δ nse3-R254E* double mutant, suggesting that it is synthetically lethal.

Furthermore, SPT analysis of Nse4–mEos3 in *nse6Δ brc1Δ* cells did not lead to further reduction in the fraction of bound complexes (*Figure 5D*).

Previous ChIP experiments have shown that Smc5/6 is enriched at repetitive genomic loci following MMS treatment and that this is dependent on Brc1 and Nse6 (*Oravcová et al., 2018*). We tested whether we could detect increased Nse4 chromatin association in response to MMS treatment in *brc1Δ* and *nse6Δ* cells. Both *brc1Δ* and *nse6Δ* cells failed to show any increase above levels detected in untreated cells upon acute exposure to MMS (*Figure 5C*), supporting the hypothesis that both Brc1 and Nse6 are required for Smc5/6 recruitment to sites of DNA damage (*Oravcová et al., 2018*).

## The Nse5–Nse6 subcomplex displays different kinetics than the Smc5/6 core complex

Intrigued by the significant role of Nse6 even in the absence of DNA damage, we investigated the dynamics of the Nse5–Nse6 complex. We tagged both Nse5 and Nse6 with mEos3 (*Figure 6—figure supplement 1*) and compared their behaviour to Nse4. In contrast to Nse2 and Smc6, which show similar chromatin association to Nse4 (*Figure 1—figure supplement 4*), both Nse5 and Nse6 displayed a broader range of displacements and were subsequently less chromatin associated (*Figure 6A,B*). This suggests Nse6 is more dynamic than other subunits and may indicate its association with the core Smc5/6 complex is transient. To determine whether chromatin association of Nse5–Nse6 is affected by that of the core complex, we introduced the *nse6-mEos3* allele into a *smc6-74* or *smc6-X* genetic background. We predicted that if Nse5–Nse6 was tightly associated with the core complex, then it would display reduced association in a *smc6-74* strain as seen with Nse4, but not in *smc6-X* (*Figures 3E* and *4B*). Tracking of Nse6–mEos3 in both mutants revealed no significant change in the fraction bound (*Figure 6D*), suggesting Nse5–Nse6 has different chromatin association dynamics to the core Smc5/6 complex. This would be indicative of Nse5–Nse6 acting to transiently stabilise or load Smc5/6 complexes on the chromatin.

## Discussion

The Smc5/6 complex is best known as a component of the DNA repair machinery that ensures the fidelity of homologous recombination (HR). However, the complex is essential in yeast, which suggests that it possesses additional functions beyond HR as deletions of core HR factors are viable (*Aragón, 2018*). The recruitment of Smc5/6 to DNA and ATP binding/hydrolysis at both the ATP sites are thought to be essential for each of its cellular roles. Understanding the molecular details of how Smc5/6 associates with DNA and/or chromatin is therefore an important step in elucidating how Smc5/6 regulates recombination and other potential DNA transactions. Here, we have established SPT as a method to probe Smc5/6 dynamics in live cells, and coupled with yeast genetics and structural studies, we uncover the key requirements for its association with chromatin.

## Smc5/6 complex features required for stable chromatin association

The Smc5/6 complex contains two separate ATP binding and hydrolysis sites. Both are formed when the Smc5 and Smc6 head domains interact. In common with all SMC complexes, the ATP binding pockets have an arginine finger, which is proposed to regulate DNA-dependent ATP hydrolysis. We show that mutating either of the Smc5 or Smc6 arginine fingers resulted in an increase in sensitivity to DNA damage and replication stress. This correlated with decreases in the fraction of bound Smc5/6 detected in SPT experiments. Interestingly, Smc5 and Smc6 arginine fingers were not equivalent as we uncovered an underlying asymmetry in the requirement of the two ATP binding sites for stable chromatin association. This asymmetry is in line with observations made for cohesin and condensin (*Elbatsh et al., 2019*; *Hassler et al., 2019*).

One of the original *smc6* mutants, *smc6-74* (A151T), maps to the residue adjacent to the arginine residue in the arginine finger domain, suggesting it is compromised in ATP hydrolysis. Using a structural model based on the *Pyrococcus furiosus* SMC head domain, we engineered a series of structurally informed mutations designed to compromise the arginine finger to various degrees. This allowed us to dial in sensitivity to DNA damaging agents that robustly correlated with a reduced ability of Smc5/6 to associate with chromatin. Taken together, these observations strongly suggest

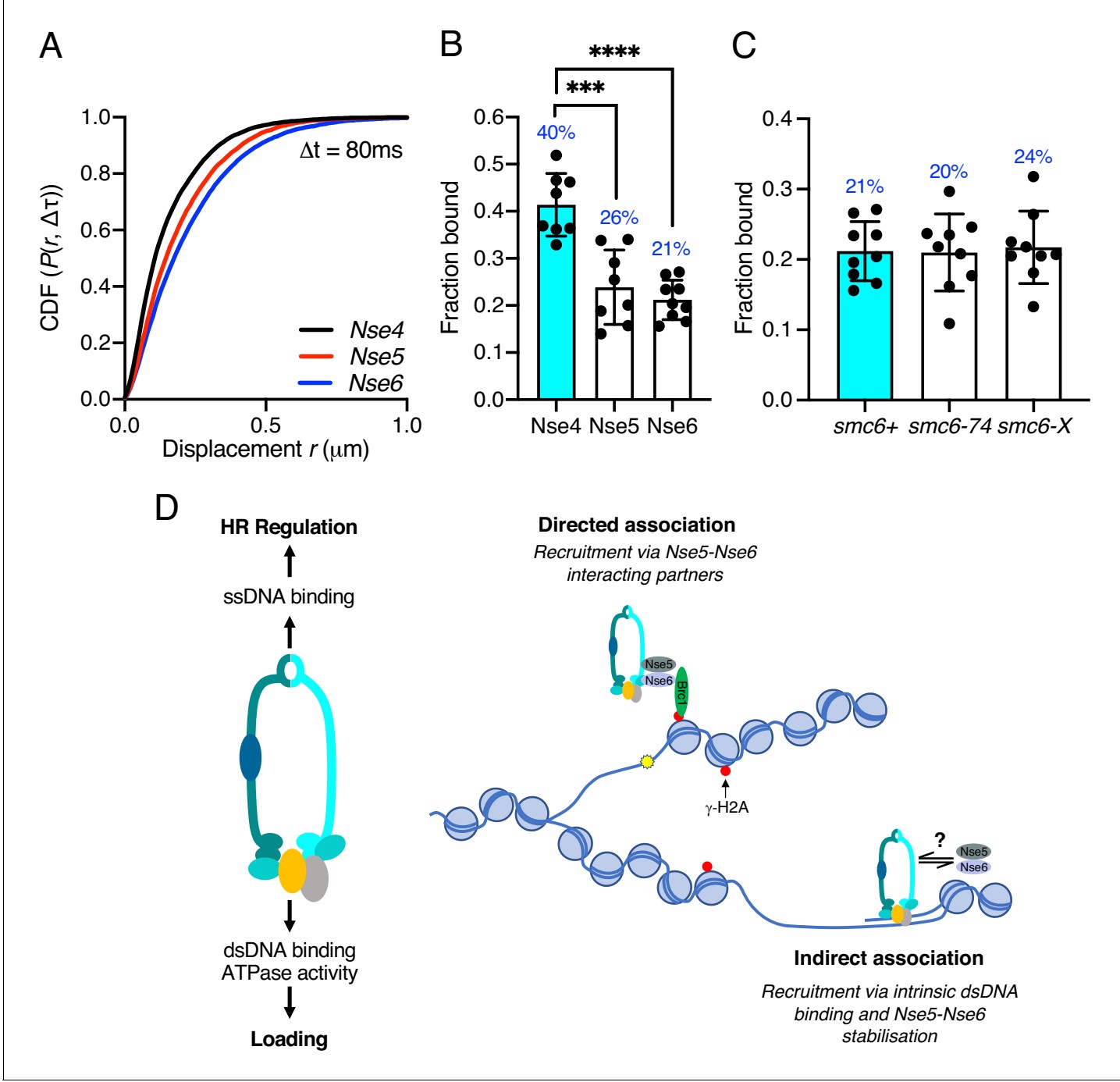

**Figure 6.** Nse5–Nse6 chromatin association is distinct from other Smc5/6 subunits. (**A**) CDF histogram of pooled single-molecule displacements at Δt = 80 ms time interval of Nse4–mEos3, Nse5–mEos3, and Nse6–mEos3. (**B**) Fraction of bound molecules extracted from Spot-On model fits from experiment in (**A**). Mean ± S.D. Black dots denote independent technical repeats, percentages denote fraction-bound value from fitting pooled data from all repeats. ***p=0.0003, ****p<0.0001. (**C**) Fraction of bound molecules extracted from Spot-On model fits from SPT Nse6–mEos3 in *smc6-74* or *smc6-X* genetic backgrounds compared to wild-type data in (**B**). (**D**) Schematic diagram of Smc5/6 DNA interactions and their roles (left) and proposed model of Smc5/6 chromatin association (right). Loading requires dsDNA binding by Nse3 and Smc5 and Smc6 ATPase activity. ssDNA binding at the hinge is not required for loading but is required for subsequent functions to regulate homologous recombination, suppress non-allelic recombination and gross chromosomal rearrangements (GCRs). Smc5/6 association with chromatin is dependent on Nse5 and Nse6 and either directed (e.g. Brc1-dependent recruitment to γ-H2A) (top) or non-directed via dsDNA binding and subsequent loading (bottom). Nse5/6 is required in both instances and may act either to directly load Smc5/6 or may stabilise its association after initial loading by dsDNA interaction.

The online version of this article includes the following figure supplement(s) for figure 6:

*Figure 6 continued on next page*

*Figure 6 continued*

**Figure supplement 1.** Phenotypic characterisation of mEos3 tagged Nse5 and Nse6 subunits.

that ATPase activity stimulated by DNA binding is pre-requisite for Smc5/6 complex DNA/chromatin association and function.

Recent structural and biophysical data for the ssDNA-binding activity of the Smc5/6 hinge domain (*Alt et al., 2017*) and the dsDNA-binding Nse1/3/4 module (*Zabrady et al., 2016*) allowed an investigation of the role for each of these two functions in promoting Smc5/6 chromatin association. The introduction of defined mutations into fission yeast demonstrated that dsDNA binding by Nse3 is required for DNA/chromatin association of the Smc5/6 complex, whereas the ability to bind ssDNA at the hinge is dispensable. Since ssDNA-binding mutants are sensitive to a range of genotoxic agents (*Alt et al., 2017*), we therefore predicted that ssDNA binding most likely plays a role in downstream processes once the complex has initially bound to dsDNA/chromatin. This would be an analogous situation to cohesin, whereby after initial DNA binding to dsDNA, capture of a second DNA moiety is only achievable for ssDNA (*Murayama et al., 2018*). This prediction is supported by results from our site-specific replication stall experiments, which indicate that increased levels of ectopic recombination occur in Smc5/6 mutants that lack the ability to interact with ssDNA correctly. This is much higher than in mutants that fail to stimulate ATPase activity and do not correctly associate with chromatin.

## Interacting factors influencing Smc5/6 chromatin association

Both Brc1 and Nse6 have been implicated in recruiting Smc5/6 to regions of γ-H2A at stalled/collapsed replication forks in fission yeast (*Oravcová et al., 2018*). We demonstrate here that deletion of either one of these factors reduces the in vivo levels of chromatin-associated Smc5/6, in both unchallenged cells and cells after exposure to MMS. Interestingly, deletion of *brc1* or preventing histone H2A phosphorylation did not generate as severe a defect in chromatin association as deletion of *nse6*. This is in agreement with recent ChIP experiments performed at discreet genomic loci (*Oravcová et al., 2018*) and demonstrates that there is at least one alternative Brc1-independent pathway for recruitment of Smc5/6 to chromatin.

To explain the data, we consider two possible modes of chromatin association: directed and non-directed association (*Figure 6C*). Directed association occurs when the complex is recruited to discrete genomic loci via interaction between the Nse5/6 subcomplex and chromatin-associated factors. This occurs via Brc1 at sites of γ-H2A, but alternative Nse5/6-interacting partners may exist to bring the complex to specific DNA structures, including stalled replication forks, HR intermediates, and double-strand breaks.

Association with the chromatin may also occur in a non-directed manner via Smc5/6's intrinsic ability to associate with DNA through the dsDNA binding site of Nse3. In this scenario, Smc5/6 initially binds DNA structures directly via Nse3 and the Nse5/Nse6 subcomplex acts transiently to stabilise this interaction. This would help explain some important observations. Firstly, while Smc5, Smc6, and Nse1-4 are all essential proteins, fission yeast cells can survive without Nse5/Nse6. In the absence of Nse5/Nse6, the complex still possesses dsDNA binding activity, but the association with the chromatin is unstable. Secondly, deletion of *nse6* is synthetically lethal with the hypomorphic dsDNA binding mutant *nse3-R254E*, suggesting the dsDNA binding activity is sufficient to retain viability in the absence of exogenous DNA damage or replicative stress. If Nse5/6 is required to stabilise DNA/chromatin association after an initial recruitment by dsDNA binding, it would explain both the essential nature of the dsDNA binding activity of Nse3 and the observations that dsDNA binding site is tightly linked to chromatin association.

These two modes are not mutually exclusive, and, in both cases, the Nse5/6 heterodimer may be acting transiently to regulate structural configurations of the complex that promote stable association with the chromatin ('loading'), much like the model for Mis4-Ssl3 being the loader for cohesin (*Ocampo-Hafalla and Uhlmann, 2011*; *Furuya et al., 1998*). Our SPT experiments show that Nse5 and Nse6 are more mobile than components of the core Smc5/6 complex suggesting alternative kinetics. This would be analogous to the cohesin-loader Scc2, which displays different dynamics to the cohesin complex and 'hops' between chromatin-bound cohesin molecules (*Rhodes et al., 2017*).

Intriguingly, two recent studies have demonstrated that Nse5/6 negatively regulates the ATPase activity of Smc5/6 in vitro, and binding to the core complex causes conformational alterations (*Hallett, 2021*; *Steigenberger et al., 2021*). Taken together with our observations that DNA-stimulated ATPase activity is required for stable loading to the chromatin, this provides an Nse5/6-dependent mechanism by which ATPase activity is repressed until a DNA substrate is encountered. We predict that once Nse5/6 inhibition of Smc5/6 ATPase is relieved, it is then released from the core complex.

In summary, by conducting a detailed characterisation of Smc5/6 chromatin association in live cells, we demonstrate that SPT is a powerful approach for studying this enigmatic complex. This methodology, when coupled with structure-led mutational analysis and yeast genetics, has provided new insights into Smc5/6 behaviour as well as clarifying previous observations from past genetic and molecular genetic experiments.

## Materials and methods

### Key resources table

| Reagent type (species) or resource | Designation | Source or reference | Identifiers | Additional information |
|---|---|---|---|---|
| Chemical compound, drug | Methylmethane sulfonate | Sigma–Aldrich | 129925–25G | |
| Chemical compound, drug | Hydroxyurea | Sigma–Aldrich | H8627-100G | |
| Chemical compound, drug | 5-Fluoroorotic acid | Formedium | 5FOA10 | |
| Other | Agarose, Type I-A, low EEO | Sigma–Aldrich | A0169-25G | |
| Other | Circular coverslips: #1.5H, ∅25 mm | Thorlabs | CG15XH | |
| Other | UV-Ozone cleaning system | Novascan | PSD-UV | |
| Software, algorithm | GDSC SMLM | Fiji plugin update site | GDSC SMLM2 | Underlying source-code is freely avaliable at https://github.com/aherbert/gdsc-smlm |
| Software, algorithm | Prism 9 | Graphpad software | | |

### *S. pombe* strain construction

*S. pombe* strains were constructed using Cre-lox-mediated cassette exchange (RMCE) as previously described (*Watson et al., 2008*). Strains were created with either essential gene replacement base strains or C-terminal tagging base strains (*Supplementary file 2*). C-terminal base strains were transformed with plasmid pAW8-mEos3.2-KanMX6 to introduce the mEos3.2 tag at the C-terminal end of the gene.

### Microscopy sample preparation

*S. pombe* cultures were grown to mid-log phase at 30°C in Edinburgh minimal media (EMM) supplemented with leucine, uracil, and adenine. Cells were harvested and washed once in phosphate-buffered saline (PBS). Cells were then resuspended in PBS, and 10 µl was deposited on an EMM-agarose pad before being mounted on ozone-cleaned circular coverslips (Thorlabs, #1.5H, ∅25 mm) and placed in a metal cell chamber for imaging (Attofluor, ThermoFisher). For replicative stress experiments, MMS was added to cultures at a final concentration of 0.03% and incubated for 5 hr before being processed for imaging.

### PALM microscopy

Live *S. pombe* cells were imaged with a custom-built microscope similar to that previously described (*Etheridge et al., 2014*). The microscope is built around an inverted Olympus IX73 body fitted with a motorised stage (Prior H117E1I4) and a heated incubation chamber (Digital Pixel Ltd). Cells were illuminated using a 561 nm imaging laser (Cobolt, Jive) and a 405 nm activation laser (LaserBoxx, Oxxius). Both laser beams were expanded and collimated and were focused to the back focal plane of an apochromatic 1.45 NA, 60 × TIRF objective (Olympus, UIS2 APON 60 × OTIRF). Both beams were angled in a highly inclined near-TIRF manner to achieve high signal-to-background. Illumination

of the sample was controlled via mechanical shutters, and all components were computer-controlled using the Micro-Manager software. The emission fluorescence from the sample was filtered with a band-pass filter (Semrock 593/40) before being expanded to create an optimised image pixel size of 101 nm after projection onto the EMCCD camera (Photometrics Evolve 512 Delta).

Samples were mounted on microscope stage and incubated at 30°C. Cells were illuminated with continuous 561 nm excitation (8.3 mW at rear aperture of objective lens) and pulsed with 100 ms 405 nm laser illumination every 10 s in order to photoconvert mEos3.2 molecules (maximum 0.23 mW at rear aperture of objective lens). We established the number of nuclei that needed to be assayed for reproducibility empirically. To ensure that single-molecule traces were recorded from a sufficient number of nuclei (>50), each biological repeat consisted of data collection from at least two separate fields of view imaged one after the other (technical repeats). Each acquisition consisted of 20,000 frames with a camera exposure time of 20 ms.

## SPT data analysis

Raw PALM data was analysed using the 'PeakFit' plugin of the GDSC single-molecule localisation microscopy plugin for Fiji 'GDSC SMLM2' (*Schindelin et al., 2012*). Single molecules were identified and localised using a 2D gaussian fitting routine (configuration file available on request). Nuclear localisations consisting of a minimum of 20 photons and localised to a precision of 40 nm or better were retained for further analysis. Single molecules were then tracked through time using the 'Trace Diffusion' GDSC SMLM plugin. Localisations appearing in consecutive frames within a threshold distance of 800 nm were joined together into a trajectory (*Etheridge et al., 2014*). Single-molecule trajectories were then exported into .csv Spot-On format using the 'Trace Exporter' plugin.

Track data was uploaded into the Spot-On web interface and was analysed using the following jump length distribution parameters: bin width (μm) = 0.01, number of timepoints = 5, jumps to consider = 4, maximum jump (μm) = 3. For all Smc5/6 components, data sets were fit with a three-state Spot-On model using the default parameters, except for $D_{slow}$min = 0.08, localisation error fit from data = yes, dZ (μm) = 0.9. The decision on which Spot-On model to fit was based on the Akaike information criterion (AIC) reported by Spot-On (see *Figure 1—figure supplement 3*). It is not clear whether this third state describes transient interactions with chromatin or arises from anomalous diffusion as a result of a crowded molecular environment (*Woringer et al., 2020*). For cohesin data sets, we fit a two-state model with the same parameters, excluding $D_{slow}$. In all cases, the model was fit to the cumulative distribution function (CDF).

Probability density function histograms and model fit were created using data combined from all three repeats of an experiment and exported from Spot-On before being graphed in Prism (Graph-Pad). Bar charts were produced by fitting data collected in each repeat (three fields of view) and extracting the fraction of bound molecules. Black circles represent the value derived for each repeat, bars represent the mean, and error bars denote standard error of the mean. Two-tailed t-test was performed in Prism software of the Spot-On $F_{bound}$ values from three repeats. Nuclear single-molecule traces used for analysis in Spot-On are available via the Open Science Framework (osf.io/myxtr).

## Structural modelling

Sequence-threaded homology models for the head domains of both *S. pombe* Smc5 and Smc6 were generated using the PHYRE2 web portal (*Kelley et al., 2015*). The potential effects of introducing single-point mutations were assessed using PyMOL (v2.32, The PyMOL Molecular Graphics System, Schrödinger, LLC).

## Yeast spot test assay

Yeast strains were cultured in yeast extract (YE) overnight to mid-log phase. Cells were harvested and resuspended to a concentration of $10^7$ cells/ml. Serial dilutions were then spotted onto YE agar plates containing the indicated genotoxic agent.

## Yeast gross chromosomal rearrangement assay

The rate of *ura4⁺* in the *RuraR* system was measured using a previously described fluctuation test (*Lambert et al., 2005*). Colonies growing on YNBA plates lacking uracil (and containing thiamine) were re-streaked onto YNBA plates containing uracil, either in the presence or in the absence of

thiamine. After 5 days, five colonies were picked from either condition, and each was grown to saturation (~48 hr) in 10 ml liquid EMM culture containing uracil, with or without thiamine.

Each culture was counted, and about $1 \times 10^7$ cells were plated in triplicate on YEA plates containing 5′-fluoroorotic acid (5′-FOA; Formedium). One hundred microliters of a 1:20,000 dilution of each saturated culture (about 200 cells) was plated in duplicate on YEA as titre plates. After 5–7 days of growth, 5-FOA-resistant colonies and colonies on YEA were counted. A proportion of 5-FOA-resistant colonies were streaked on YNBA lacking uracil to verify *ura4* gene function loss. These *ura4*⁻ colonies were used in the translocation PCR assay as described previously (*Lambert et al., 2005*). The rate of *ura4* loss per cell per generation was calculated using the maximum likelihood estimate of the Luria-Delbruck with a correction for inefficient plating (*Zheng, 2008*). We performed all computations using the R package rSalvador (*Zheng, 2017*).

## Acknowledgements

We would like to acknowledge Anders Hansen and Maxime Woringer for their help with initial implementation of the Spot-On software for the analysis of SPT data in fission yeast. We thank Sarah Lambert for control strains for the GCR assay and Steven F Lee for single-molecule microscopy advice. We also thank J Palecek for the *nse3-R254E* strain. AMC acknowledges support from the Wellcome trust (110047/Z/15/Z), and JMM and AWO acknowledge support from the MRC (MR/P018955/1). TJE would like to dedicate this manuscript to the memory of Vivien Horobin.

## Additional information

### Funding

| Funder | Grant reference number | Author |
| --- | --- | --- |
| Wellcome Trust | 110047/Z/15/Z | Antony M Carr |
| Medical Research Council | MR/P018955/1 | Antony W Oliver<br>Johanne M Murray |

The funders had no role in study design, data collection and interpretation, or the decision to submit the work for publication.

### Author contributions

Thomas J Etheridge, Conceptualization, Formal analysis, Investigation, Methodology, Writing - original draft, Writing - review and editing, Designed and constructed custom microscope; Desiree Villahermosa, Formal analysis, Investigation; Eduard Campillo-Funollet, Software, Formal analysis; Alex David Herbert, Software, Formal analysis, Investigation, Wrote and benchmarked the GDSC SMLM custom single-molecule software plugin; Anja Irmisch, Adam T Watson, Hung Q Dang, Investigation; Mark A Osborne, Resources, Methodology, Designed and constructed custom microscope; Antony W Oliver, Conceptualization, Formal analysis, Funding acquisition, Investigation, Writing - original draft, Writing - review and editing, Performed structural analysis and designed mutations; Antony M Carr, Conceptualization, Funding acquisition, Writing - original draft, Project administration, Writing - review and editing; Johanne M Murray, Conceptualization, Supervision, Funding acquisition, Writing - original draft, Project administration, Writing - review and editing

### Author ORCIDs

Thomas J Etheridge (iD) https://orcid.org/0000-0001-8144-6917
Eduard Campillo-Funollet (iD) http://orcid.org/0000-0001-7021-1610
Alex David Herbert (iD) http://orcid.org/0000-0002-9843-9980
Hung Q Dang (iD) http://orcid.org/0000-0002-1226-0235
Antony W Oliver (iD) http://orcid.org/0000-0002-2912-8273
Antony M Carr (iD) http://orcid.org/0000-0002-2028-2389
Johanne M Murray (iD) https://orcid.org/0000-0001-9225-6289

Decision letter and Author response
Decision letter https://doi.org/10.7554/eLife.68579.sa1
Author response https://doi.org/10.7554/eLife.68579.sa2

## Additional files

### Supplementary files

• Supplementary file 1. Fluctuation experiment data tables. Data from individual experimental repeats of *ura4* loss assay in *Figure 4C*.

• Supplementary file 2. Strain table. Strains used during this study.

• Transparent reporting form

### Data availability

Single molecule traces exported from GDSC SMLM plugin and used for analysis in SpotOn software are available via the Open Science Framework (http://osf.io/myxtr).

The following dataset was generated:

| Author(s) | Year | Dataset title | Dataset URL | Database and Identifier |
|---|---|---|---|---|
| Etheridge TJ | 2020 | Single-molecule live cell imaging of the Smc5/6 DNA repair complex | https://osf.io/myxtr/ | Open Science Framework, myxtr |

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
