## [Decision Letter]

**Acceptance summary:**

The authors use the photoconvertible fluorophore mEos3 fused to Sm5/6 complex components to study the dynamics of these complexes in live fission yeast cells at single molecule resolution by single-particle tracking photo-activated localization microscopy (sptPALM). Specific point mutations affecting the dsDNA binding, ssDNA binding or ATPase activity of the SMC5/6 complex or the interacting factors Brc1 and Nse6 were analyzed by sptPALM and genetic assays to measure DNA damage survival, mutation rates, and genomic rearrangements at stalled replication forks. The authors propose a model of two modes of chromatin association of the Smc5/6 complex by direct binding to dsDNA and indirect recruitment through Brc1 bound to γ H2A. Moreover, the authors conclude that ssDNA binding is not required for Smc5/6 chromatin association but for a more downstream action in maintaining genomic stability by homologous recombination.

**Decision letter after peer review:**

[Editors’ note: the authors submitted for reconsideration following the decision after peer review. What follows is the decision letter after the first round of review.]

Thank you for submitting your work entitled "Single-molecule live cell imaging of the Smc5/6 DNA repair complex" for consideration by *eLife*. Your article has been reviewed by 3 peer reviewers, including Wolf-Dietrich Heyer as the Reviewing Editor and Reviewer #1, and the evaluation has been overseen by a Senior Editor.

Our decision has been reached after consultation between the reviewers. Based on these discussions and the individual reviews below, we regret to inform you that your work will not be considered further for publication in *eLife*.

In its current form, the manuscript might be better suited for a more specialized methods paper, but the impact of the work could be significantly raised by addressing the points #1-3 discussed below. As the requested additions are too extensive for a revision, we would invite a new submission of a revised manuscript with substantial new experiments to address these points, if you elect this course of action. We would make every effort to have the same review team evaluate the new manuscript.

Summary:

The authors use the photoconvertible fluorophore mEos3 fused to cohesin, condensin, and Sm5/6 complex components to study the dynamics of these complexes in live fission yeast cells at single molecule resolution by single-particle tracking photo-activated localization microscopy (sptPALM). Specifically, the authors find that Brc1 and Nse6 both promote chromatin loading of Smc5/6 but with brc1∆ having a less dramatic defect than nse6∆. The role of Brc1 is consistent with a recent report by (Oravcova et al., 2019). Consistent with Brc1 being recruited to chromatin by binding γ-H2A (Williams et al., 2010), brc1∆ and hta-SA mutants exhibit comparable defects in chromatin association of Smc5/6. Next, the authors show that mutation of each of the two ATP binding sites cause DNA damage sensitivity and defective chromatin association. Importantly, the authors find that dsDNA but not ssDNA binding is required for loading Smc5/6 on chromatin but not for prevention of gross chromosomal rearrangements.

Overall, the manuscript is well written and the data of high quality, but it does not advance the field much in terms of our understanding of the role of Smc5/6 in DNA repair. Moreover, the sptPALM method is not compared sufficiently with existing methods (ChIP and cytological analysis of foci), which appear to have a better dynamic range that sptPALM, to allow readers to choose between these methods for their experiments. Moreover, there are a number of technical issues that need clarification and resolution.

The key points to be addressed in case the authors elect to re-submit the manuscript:

1. This manuscript represents the first time sptPAML is used to derive the dynamic behavior of the Smc5/6 complex in live cells. There are a number of technical concerns that should be addressed.

• Lines 119/120: The authors state that DNA-bound proteins have constrained diffusion, which is clearly correct. The question is whether all constrained diffusion is caused by DNA binding, or whether binding to other stationary nuclear structures, such as the nuclear pore complex, or confinement in phase-separated compartments, could also lead to constraints?

• Line 162 ff: The authors document significant differences between cohesin and Smc5/6 mobility and mention ChIP experiments that suggest co-localization. This discrepancy should be examined and discussed more than is done on lines 374/5?

• What does it mean to be 50% chromatin bound be the sptPALM method? Does FRAP (or chromatin/soluble fractionation) also show a similar immobile fraction?

• The higher displacement value observed for Nse6-mEos3 compared with Nse4 could be due to the tag differentially affecting different subunits, particularly given that Figure S1 showed that cells containing Nse6-mEos3, but not Nse4-mEos3, are sensitive to MMS.

• Could the different displacement values of Nse4-mEos3 between brc1∆ vs nse6∆ cells be due to the mutant cells accumulating at different cell cycle phases given the slow growth of the mutants?

• Could the different mutants tested affect Nse4-mEos3 protein levels thus may affect data?

• Could the nse3-R254E or hinge mutants tested affect the Smc5/6 complex stability, thus affecting the signals?

• Figure 1B, the result raised the question whether the Rad21-mEos3 expression level is greatly reduced compared to the endogenous Rad21. Does mEos3 fusion affect the protein levels of the examined Smc5/6 subunits? Clarification of protein levels of the fusion proteins is important for this work, as the sptPAML data can be affected by protein levels.

• Figure 1C, it is unclear how cells in mitosis vs. interphase were assigned. Also, what are criteria for determining nucleus vs cytoplasm in these images?

• Judging from Figure 2C and S1B, Nse4 and Rad21 appear to be present mostly in large bright clusters, though significant amounts of weaker signals are also visible, particularly for Nse4. This could suggest that two populations of proteins (within and outside the clusters) exist. Do data in Figure 2A and B represent only the signals within the cluster or exhibiting weaker signals, or both (averaged).

2. What is the functional significance of the observed asymmetry in use of the two ATP binding sites? Moreover, the conclusion that "DNA-dependent ATP hydrolysis by the two binding pockets is not equivalent" could be strengthened by a direct assessment of the ATPase activity of the Smc5 (smc5-R77A) or Smc6 (smc6-R150A).

3. Why does suppression of gross chromosomal rearrangements require ssDNA binding but apparently not chromatin binding? The authors propose a model for HRR by Smc5/6 where dsDNA binding happens first and independent of DNA damage and ssDNA binding happens during DNA repair. It would strengthen the impact of the manuscript to test this model. For example: can DNA damage sensitivity and the chromatin association defect of dsDNA binding nse3-R254E mutant be rescue by fusion of Nse4 or Nse6 to Brc1? What distinguishes resistance to replication stress and suppression of gross chromosomal rearrangements?

Additional points for the authors to consider:

4. Figure 3 C, D: The results beg the question of the phenotype of the brc1 nse6 double mutant. This could be very informative to start addressing the models developed in the discussion.

5. Figure 3E: The treatment with MMS is rather long (5 hours). How was this time chosen? Likely DNA lesions would have been processed and perhaps even repaired and some cells dead at 5 hours. I therefore suggest to do a time course for Smc5/6 chromatin association. The kinetics of chromatin association might simply be different in the brc1∆ and nse6∆ mutants.

6. Figure 3F: The model in A predicts epistasis between the brc1 and the γ H2AX mutants, which should be tested here or has this been tested before?

7. Figure 5, the Alt 2017 paper suggests that while smc5-R609, 615E abolishes ssDNA binding, smc5-Y612G or smc6-x, and to a large extent Smc6-F528A, exhibit close to wild-type level of ssDNA binding. Here, these mutants showed similar Nse4 sptPALM behavior. Thus, these observations would argue that features other than ssDNA binding affect Nse4-chromatin association.

8. Figure 6B: The fluctuation tests lack error bars. Since Figure S6 is labeled as a repeat of Figure 6B left, I assume these are single determinations. The data should be reported as mean of n=3. The rate difference between Figure 6B left and Figure S6 are unusually large for a rate determination, which is usually quite robust. All the more a reason to bolster these data with n=3.

Text changes or clarifications:

9. Column graphs should be converted to scatter "superplots" as recently proposed (Lord et al., 2020) to show individual cell data and variability between experiments.

10. Line 35: insert 'function of the' in front of Smc5/6.

11. Line 42: It might help the reader to include the statement here, which is found in the beginning of the discussion (lines 355 ff), to indicate that SMC must have functions beyond HR.

12. Line 129: What is the evidence that the single particle of condensin contains a single condensin complex?

13. Page 10: Please provide a reference for the epistatic relationship between smc6-X (R706C) and smc6-74 (A151T) and rad51-d in response to MMS.

14. Figure 3A: The evidence that Brc1 is binding to gammaH2AX should be discussed.

15. Figure 5A: Please indicate the gene/mutations used in the scheme.

References:

Alt, A, Dang HQ, Wells OS, Polo LM, Smith MA, McGregor GA, Welte T, Lehmann AR, Pearl LH, Murray JM, Oliver AW. (2017) Specialized interfaces of Smc5/6 control hinge stability and DNA association. Nature Comm 8, 14011

Lord, S.J., Velle, K.B., Mullins, R.D., and Fritz-Laylin, L.K. (2020). SuperPlots: Communicating reproducibility and variability in cell biology. J Cell Biol 219.

Oravcova, M., Gadaleta, M.C., Nie, M., Reubens, M.C., Limbo, O., Russell, P., and Boddy, M.N. (2019). Brc1 Promotes the Focal Accumulation and SUMO Ligase Activity of Smc5-Smc6 during Replication Stress. Mol Cell Biol 39.

Williams, J.S., Williams, R.S., Dovey, C.L., Guenther, G., Tainer, J.A., and Russell, P. (2010). gammaH2A binds Brc1 to maintain genome integrity during S-phase. EMBO J 29, 1136-1148.

Reviewer #1:

The authors use the photoconvertible fluorophore mEos3 fussed to cohesin, condensin, and Sm5/6 complex components to study the dynamics of these complexes in live fission yeast cells at single molecule resolution by single-particle tracking photo-activated localization microscopy (PALM). Using the 'spot-on' program to analyze single particle tracts, the authors define differences between the three complexes in their nuclear dynamics, determine the effect of DNA binding and ATPase mutants in the Smc5/6 complex on constrained diffusion, which is interpreted a reporting on chromatin association. In addition, they define two distinguishable recruitment pathways for Smc5/6 complex. Overall, this is a novel and fruitful approach to study the enigmatic Smc5/6 complex, which generated significant novel insights not previously accessible. The authors should consider the following points in a revision.

1. Lines 119/120: The authors state that DNA-bound proteins have constrained diffusion, which is clearly correct. The question is whether all constrained diffusion is caused by DNA binding, or whether binding to other stationary nuclear structures, such as the nuclear pore complex, or confinement in phase-separated compartments, could also lead to constraints?

2. Line 162 ff: The authors document significant differences between cohesin and Smc5/6 mobility and mention ChIP experiments that suggest co-localization. Maybe this discrepancy could be discussed more than is done on lines 374/5?

3. Figure 3 C, D: The results beg the question of the phenotype of the brc1 nse6 double mutant. I think this could be very informative to start addressing the models developed in the discussion.

4. Figure 3F: The model in A predicts epistasis between the brc1 and the γ H2AX mutants, which should be tested here.

5. Figure 6B: The fluctuation tests lack error bars. Since Figure S6 is labeled as a repeat of Figure 6B left, I assume these are single determinations. The data should be reported as mean of n=3. The rate difference between Figure 6B left and Figure S6 are unusually large for a rate determination, which is usually quite robust. All the more a reason to bolster these data with n=3.

Additional points:

6. Line 35: insert 'function of the' in front of Smc5/6.

7. Line 42: It might help the reader to include the statement here, which is found in the beginning of the discussion (lines 355 ff), to indicate that SMC must have functions beyond HR.

8. Line 129: What is the evidence that the single particle of condensin contains a single condensin complex?

9. Figure 3A: The evidence that Brc1 is binding to gammaH2AX should be discussed.

10. Figure 5A: Please indicate gene/mutations used in the scheme.

Reviewer #2:

The manuscript entitled "Single-molecule live cell imaging of the Smc5/6 DNA repair complex" by Etheridge and colleagues introduces single-particle tracking PALM (sptPALM) microscopy to study chromatin association of the Smc5/6 complex. The authors use this technique in combination with genetics to dissect the chromatin recruitment of the Smc5/6 in unchallenged cells and after genotoxic stress. Specifically, the authors find that Brc1 and Nse6 both promote chromatin loading of Smc5/6 but with brc1∆ having a less dramatic defect than nse6∆. The role of Brc1 is consistent with a recent report by (Oravcova et al., 2019). Consistent with Brc1 being recruited to chromatin by binding γ-H2A (Williams et al., 2010), brc1∆ and hta-SA mutants exhibit comparable defects in chromatin association of Smc5/6. Next, the authors show that mutation of each of the two ATP binding sites cause DNA damage sensitivity and defective chromatin association. Importantly, the authors find that dsDNA but not ssDNA binding is required for loading Smc5/6 on chromatin but not for prevention of gross chromosomal rearrangements. The manuscript is well written and the data of high quality, but it does not advance the field much in terms of our understanding of the role of Smc5/6 in DNA repair. Moreover, the sptPALM method is not compared sufficiently with existing methods (ChIP and cytological analysis of foci), which appear to have a better dynamic range that sptPALM, to allow readers to choose between these methods for their experiments. In its current form, the manuscript might be better suited for a more specialized methods paper, but the impact of the work could be significantly raised by addressing some of the key questions raised by the study such as:

What is the functional significance of the observed asymmetry in use of the two ATP binding sites?

Why does suppression of gross chromosomal rearrangements require ssDNA binding but apparently not chromatin binding?

Additional major issues to address:

1. Page 8: The conclusion that "DNA-dependent ATP hydrolysis by the two binding pockets is not equivalent" would be strengthen by a direct assessment of the ATPase activity of the Smc5 (smc5-R77A) or Smc6 (smc6-R150A).

2. The authors propose a model for HRR by Smc5/6 where dsDNA binding happens first and independent of DNA damage and ssDNA binding happens during DNA repair. It would strengthen the impact of the manuscript to test this model. For example: can DNA damage sensitivity and the chromatin association defect of dsDNA binding nse3-R254E mutant be rescue by fusion of Nse4 or Nse6 to Brc1? What distinguishes resistance to replication stress and suppression of gross chromosomal rearrangements?

3. What does it mean to be 50% chromatin bound be the sptPALM method? Does FRAP (or chromatin/soluble fractionation) also show a similar immobile fraction?

4. Figure 3E: The treatment with MMS is rather long (5 hours). How was this time chosen? Likely DNA lesions would have been processed and perhaps even repaired and some cells dead at 5 hours. I therefore suggest to do a time course for Smc5/6 chromatin association. The kinetics of chromatin association might simply be different in the brc1∆ and nse6∆ mutants.

References:

Lord, S.J., Velle, K.B., Mullins, R.D., and Fritz-Laylin, L.K. (2020). SuperPlots: Communicating reproducibility and variability in cell biology. J Cell Biol 219.

Oravcova, M., Gadaleta, M.C., Nie, M., Reubens, M.C., Limbo, O., Russell, P., and Boddy, M.N. (2019). Brc1 Promotes the Focal Accumulation and SUMO Ligase Activity of Smc5-Smc6 during Replication Stress. Mol Cell Biol 39.

Williams, J.S., Williams, R.S., Dovey, C.L., Guenther, G., Tainer, J.A., and Russell, P. (2010). gammaH2A binds Brc1 to maintain genome integrity during S-phase. EMBO J 29, 1136-1148.

Reviewer #3:

The authors used sptPAML to examine the Smc5/6 subunits for their chromatin association in live fission yeast cells. They found that Smc5/6 appears to be in the nucleus throughout the cell cycle but exhibited more dynamic association with chromatin compared with cohesin. In addition, Nse6 appears to be more dynamic than Nse2, Nse4, and Smc5 or 6. The authors further examined several mutants in this assay. They found several mutants show defects of Nse4 chromatin associated and these include: 1) brc1∆ (or H2A-SA) and nse6∆ with only nse6∆ exhibit a complete defect, regardless of MMS treatment. 2) Smc5 or 6 ATP hydrolysis mutants, with Smc6 mutant exhibiting stronger effects. 3) a Nse3 mutant reduce in vitro dsDNA binding, and 4) hinge mutant defective afffecting ssDNA binding exhibiting defect only after MMS treatmen. Lastly, the authors show that the smc6-x and -74 increased rates of DNA deletion due to fork stalling (HR based events). Overall, this is an interestingly work that for the first time uses sptPAML to derive the dynamic behavior the Smc5/6 complex in live cells and using this assay to assess how different mutants of the complex can influence such behavior. While several conclusions are expected from previous reports, a few potentially significant conclusions are drawn. The work could benefit from technique clarifications and adjustment in interpretations as detailed below.

Given data are mostly acquired using sptPALM, an expert of this technique and analyses (spot-on, model fitting, PDF graphs) should be asked to assess data quality and interpretations. For example, can displacement values be strictly interpreted as reflecting chromatin-bound protein fraction; could they reflect other aggregation behavior of the proteins, such as phase separation? In addition, this reviewer felt that the interpretation of sptPALM data can be somewhat biased and the authors should consider other possibilities. For example, whether the higher displacement value observed for Nse6-mEos3 compared with Nse4 could be due to the tag differentially affecting different subunit, particularly given that Figure S1 showed that cells containing Nse6-mEos3, but not Nse4-mEos3, are sensitive to MMS. Also, could different displacement values of Nse4-mEos3 between brc1∆ vs nse6∆ cells be due to mutant cells are accumulated at different cell cycle phases given the slow growth of the mutants. Could the different mutants tested affect Nse4-mEos3 protein levels thus may affect data? Could the nse3-R254E or hinge mutants tested affect the Smc5/6 complex stability, thus affecting the signals?

Figure 1B, the result raised the question whether the Rad21-mEos3 expression level is greatly reduced than the endogenous Rad21. Along the same line, does the mEos3 fusion affect the protein levels of the examined Smc5/6 subunits? Clarification of protein levels of the fusion proteins is important for this work, as sptPAML data can be affected by protein levels.

Figure 1C, it is unclear how cells in mitosis vs. interphase were assigned. Also, what are criteria for determining nucleus vs cytosplam in these images?

Judging from Figure 2C and S1B, Nse4 and Rad21 appear to be present mostly in large bright clusters, though significantly amounts of weaker signals are also visiable, particularly for Nse4. This could suggest that two populations of proteins (within and outside the clusters). Do data in Figure 2A and B represent only the signals within the cluster or exhibiting weaker signals, or both (averaged).

Figure 5, the Alt 2017 paper suggests that while smc5-R609, 615E abolishes ssDNA binding, smc5-Y612G or smc6-x, and to a large extent Smc6-F528A, exhibit close to wild-type level of ssDNA binding. Here these mutants showed similar Nse4 sptPALM behavior. Thus, these observations would argue that features other than ssDNA binding affect Nse4-chromatin association.

[Editors’ note: further revisions were suggested prior to acceptance, as described below.]

Thank you for submitting your article "Live-cell single-molecule tracking highlights requirements for stable Smc5/6 chromatin association in vivo" for consideration by *eLife*. Your article has been reviewed by the same 3 peer reviewers as the initial submission, including Wolf-Dietrich Heyer as the Reviewing Editor and Reviewer #1, and the evaluation has been overseen by a Kevin Struhl as the Senior Editor. The reviewers have discussed their reviews with one another, and the Reviewing Editor has drafted this to help you prepare a revised submission.

Summary:

This is the resubmission of a previously rejected manuscript that was reviewed by the same reviewers as the original submission. The authors focused the manuscript on the Smc5/6 complex and the manuscript benefitted from the streamlining. The revision and the response clarified many of the technical concerns.

New experimentation with the Nse5-mEos3 fusion corroborates the data with the NSe6 fusion. Additional controls using FACS analysis exclude cell cycle differences as the cause for the observed effects in brc1 and nse6 deficient cells. Previous published work provides controls for protein levels as well as the analysis of the brc1 nse6 double mutant. New experimentation conducting a time course and additional clarification confirms that the 5 hr time point for the MMS treatment is adequate. New experimentation solidified the mutation rate data.

Essential revisions:

The authors report in their rebuttal that they have performed SPT at hourly intervals in wild type cells and found that the fraction bound was highest at 4-5 hours. These data should be included in the manuscript, maybe as a supplemental figure.

---

## [Author Response]

[Editors’ note: the authors resubmitted a revised version of the paper for consideration. What follows is the authors’ response to the first round of review.]

In its current form, the manuscript might be better suited for a more specialized methods paper, but the impact of the work could be significantly raised by addressing the points #1-3 discussed below. As the requested additions are too extensive for a revision, we would invite a new submission of a revised manuscript with substantial new experiments to address these points, if you elect this course of action. We would make every effort to have the same review team evaluate the new manuscript.

We agree that the previous manuscript focussed too much on ‘proving’ the application of the singe-molecule technique, which may not have been necessary as it is now becoming more of a standard technique. Whilst we have found that the demonstration of the single-molecule tracking approach using the SMC complex condensin is visually beneficial for conference talks, we recognised that it detracted the manuscript somewhat from its central focus, thus we have removed this data. We have re-written the text to briefly introduce the type of SPT analysis that we do in the first figure.

We believe that this study will be of interest to the wide readership of *eLife* as a significant number of single-molecule microscopy papers exploring a wide range of topics have recently been published in the journal.

Recently, the same kinetic modelling approach (‘Spot-On’ – also published in *eLife*^3^) has been used to address H2A.Z eviction by RNA Pol II^4^.

The key points to be addressed in case the authors elect to re-submit the manuscript:1. This manuscript represents the first time sptPAML is used to derive the dynamic behavior of the Smc5/6 complex in live cells. There are a number of technical concerns that should be addressed.• Lines 119/120: The authors state that DNA-bound proteins have constrained diffusion, which is clearly correct. The question is whether all constrained diffusion is caused by DNA binding, or whether binding to other stationary nuclear structures, such as the nuclear pore complex, or confinement in phase-separated compartments, could also lead to constraints?

This is an interesting question and one which is rather general to this methodology. In our study we generated a hypomorphic mutation *nse3R254E* which has been shown in vitro to significantly reduce binding to double stranded DNA. In our SPT experiments this mutant displayed >2fold reduction in the bound fraction to a value similar to the most severe ATPase mutant. Complete abrogation of dsDNA binding is lethal and thus SPT cannot be performed in such a mutant. In response to this question, we attempted to deplete the wild type copy of *nse3* whilst inducing expression of lethal *nse3* dsDNA binding mutant, however fusion of the auxin-inducible degron (AID) tag to Nse3 was itself lethal.

Whilst we believe that there could indeed be confinement in compartments of the nucleus, this attenuated diffusion would still be higher than the apparent diffusion coefficient of the ‘bound’ Nse4 fraction (*D =* 0.005m ^2^ s^-1^ ). Indeed, this may explain the slow diffusing population that was extracted from Spot-On kinetic modelling and will be investigated in a future study.

• What does it mean to be 50% chromatin bound be the sptPALM method? Does FRAP (or chromatin/soluble fractionation) also show a similar immobile fraction?

The ‘fraction bound’ value refers to the fraction of molecules that were

DNA/chromatin bound at the time of the experiment. This value is extracted using the Spot-On software which fits a kinetic model to singlemolecule displacement distributions. From this distribution it extracts different mobility states characterized by a diffusion coefficient (Free/Bound) and a fraction of the total population residing in this state (See new Figure 1C).

DNA bound molecules will have significantly attenuated diffusion and thus present trajectories that are essentially stationary and comprised of displacements roughly equal to the localisation precision of the experiment. These can be seen as the peaks in the displacement distributions such as Figure 1D.

With respect to other approaches – firstly, we and others have found that biochemical analysis of Smc5/6 chromatin association using chromatin immunoprecipitation (ChIP) or fractionation have always been limiting due to poor enrichment of the complex. This approach hides the dynamics of a process as they usually require chemical fixation – this was the main reason for initiating this study.

Secondly, SPT and FRAP are complementary approaches that allow the study of the mobility of molecules. Whilst FRAP provides ensemble information of a population of molecules, SPT is capable of visualising the movement of individual fluorescently labelled molecules. Inevitably, FRAP is somewhat limited by ensemble averaging and cannot reliably describe complex diffusion scenarios, such as mixed populations, different modes of diffusion and molecules that show sub-diffusive behaviour. Thus, SPT is a favourable methodology. We would predict that FRAP would provide similar data for wild type cells and mutants that have significant chromatin association defects – but it may not be sensitive enough to detect smaller changes such as that detected in Figures 3 and 5.

• The higher displacement value observed for Nse6-mEos3 compared with Nse4 could be due to the tag differentially affecting different subunits, particularly given that Figure S1 showed that cells containing Nse6-mEos3, but not Nse4-mEos3, are sensitive to MMS.

This is true, the Nse6-mEos3, although it shows wildtype growth in the absence of DNA damage, the strain is slightly sensitive to MMS at higher concentrations. Thus, to address this question we created Nse5-mEos3. This strain which demonstrates wild type levels of sensitivity at high MMS concentrations (Supplementary Figure 1B), and showed similar fraction bound values for the Nse6-mEos3 strain (Figure 6B). This confirms our observation that the Nse5/6 subcomplex behaves differently to the rest of the Smc5/6 complex.

• Could the different displacement values of Nse4-mEos3 between brc1∆ vs nse6∆ cells be due to the mutant cells accumulating at different cell cycle phases given the slow growth of the mutants?

Flow cytometry analysis of the DNA content of wildtype, *brc1* and *nse6* cells pre- and post-treatment with MMS showed similar profiles suggesting that reduction in chromatin association was not due to different cell cycle kinetics.

• Could the different mutants tested affect Nse4-mEos3 protein levels thus may affect data?

We know from previous work that destabilising one Smc5/6 component causes concomitant loss of expression of other subunits^5.^ The mutants used in this study have all previously been published and assessed for expression – Alt et al. 2017 in supplementary figure 3 and Zabrady et al. 2016 in supplementary figure 4^6,7^. In these studies, they found no evidence that the mutations lead to decreased protein expression level. Thus, we are confident that there is no effect on Nse4-mEos3 protein levels in our study.

• Figure 1B, the result raised the question whether the Rad21-mEos3 expression level is greatly reduced compared to the endogenous Rad21. Does mEos3 fusion affect the protein levels of the examined Smc5/6 subunits? Clarification of protein levels of the fusion proteins is important for this work, as the sptPAML data can be affected by protein levels.

The protein expression level of the cohesin kleisin subunit has been predicted to be less than that of other SMC components by quantitative mass spectrometry in fission yeast (rad21 = 173 molecules/cell, Cnd2 = 3502 molecules/cell, Nse1 = 1900 molecules/cell, Smc5 = 2329 molecules/cell)^8^. This suggests that the expression of cohesin kleisin is 10-20-fold lower than other kleisins or SMC subunits, which would support our western blot data. Furthermore, we have performed SPT analysis on an alternative cohesin subunit (Smc1 – Supplementary figure 3) and found complementary results. Our interpretation of this data, that cohesin has higher levels of chromatin association than Smc5/6 in our samples is therefore supported by this.

2. What is the functional significance of the observed asymmetry in use of the two ATP binding sites? Moreover, the conclusion that "DNA-dependent ATP hydrolysis by the two binding pockets is not equivalent" could be strengthened by a direct assessment of the ATPase activity of the Smc5 (smc5-R77A) or Smc6 (smc6-R150A).

It is currently unclear as to why there is an asymmetry in the use of the two ATP binding sites of Smc5 and Smc6. A mechanism for such asymmetry has been described for cohesin DNA release^9^ and condensin kleisin restructuring^10^. However, these studies have relied heavily on structural and biochemical data which is currently outside of the scope of this study but will be the focus of future experiments.

3. Why does suppression of gross chromosomal rearrangements require ssDNA binding but apparently not chromatin binding?

Our site-specific fork restart experiments (Figure 4C) show increased gross chromosomal rearrangements (GCR) in the *smc6-*X hinge mutant but not ATPase/dsDNA binding mutants. Restart at the RTS1 barrier requires homologous recombination, thus our data suggests that ssDNA interactions at the hinge are required to regulate these HR events.

As the ATPase and dsDNA binding mutants are hypomorphic, we predict that chromatin association of Smc5/6 is still achievable in these mutants but is significantly less efficient. Thus, it may be the case that eventually the complex is able to load at the RTS1 barrier, and efficiently direct HR repair. This is consistent with the detection of a fraction of DNA bound complex detected in SPT experiment in these mutants. In the hinge mutant, initial recruitment is not compromised (as evidenced by SPT), but defective interaction with ssDNA at HR events causes increase in GCR. These data suggest in some cases it is worse to recruit a dysfunctional Smc5/6 compared to reducing its ability to associate with the DNA.

The authors propose a model for HRR by Smc5/6 where dsDNA binding happens first and independent of DNA damage and ssDNA binding happens during DNA repair. It would strengthen the impact of the manuscript to test this model. For example: can DNA damage sensitivity and the chromatin association defect of dsDNA binding nse3-R254E mutant be rescue by fusion of Nse4 or Nse6 to Brc1?

Our model predicts that initial dsDNA binding and ATPase activity is required for stable association with the chromatin prior to its downstream functions at DNA repair events. We do not predict that this occurs independently of DNA damage. The chromatin association we detect in unchallenged cells may well be due to endogenous damage/replication fork arrest. This would be consistent with the essential nature of Smc5/6 and the observation that a proportion of this association is to regions of

γ-H2A.

We do not believe that a fusion experiment such as the one suggested would be beneficial to test this model. Firstly, stable association of the complex would still require efficient DNA binding in order to stimulate ATPase activity, increasing local concentration would not promote this. In support of this hypothesis is the fact that overexpression of Brc1 partially rescues DNA damage sensitivity of *smc6-74* (ATPase) mutants but not *smc6-X* (hinge/ssDNA interaction)^11^. This has been shown genetically to require Rad18 which channels repair into the post-replication repair pathway rather than increasing Smc5/6 loading^12^.

What distinguishes resistance to replication stress and suppression of gross chromosomal rearrangements?

This is an interesting question and one that the field cannot fully answer yet. Our prediction is that this would depend on the type of blockage that a replication fork encounters. Restart of replication forks at the RTS1 barrier used in our *RuraR*system (Figure 4C) is entirely dependent on homologous recombination. HR is also required for replication at the rDNA repeats, a process which Smc5/6 has also been linked with regulating. In such cases, we predict Smc5/6 is required to suppress gross chromosomal rearrangements caused by non-allelic HR. In other instances of replication stress, such as UV/MMS lesions, a variety of structures not seen at RTS1 may be generated. Distinguishing the role of Smc5/6 at different DNA structures is the focus of current and future research.

Additional points for the authors to consider:4. Figure 3 C, D: The results beg the question of the phenotype of the brc1 nse6 double mutant. This could be very informative to start addressing the models developed in the discussion.

This epistasis was tested in Oravcova, et al.^13.^ The authors found that the double *brc1Δ nse6Δ* mutant was more sensitive to genotoxic agents which suggests that each protein has independent functions. This is consistent with our model that places Nse5/6 as a regulator of stable chromatin association as well interacting with recruiting partners such as Brc1. In budding yeast, the Brc1 homologue Rtt107 has been shown to interact with a variety of factors other than Smc5/6, including Mms22 and Slx4, suggesting it too has independent functions^14^.

5. Figure 3E: The treatment with MMS is rather long (5 hours). How was this time chosen? Likely DNA lesions would have been processed and perhaps even repaired and some cells dead at 5 hours. I therefore suggest to do a time course for Smc5/6 chromatin association. The kinetics of chromatin association might simply be different in the brc1∆ and nse6∆ mutants.

All previous work investigating the dynamics of Smc5/6 chromatin association has relied on chromatin immunoprecipitation (ChIP) approaches. We wanted to make sure that our experiments using MMS treatment were comparable to previous studies. As we were investigating previous predictions from Oravcova, et al. 2019 we decided to use the concentration and timings from this study^13^. In response to the question, we performed SPT at hourly intervals in wild type cells and found that the fraction bound was highest at 4-5 hours.

6. Figure 3F: The model in A predicts epistasis between the brc1 and the γ H2AX mutants, which should be tested here or has this been tested before?

The *hta1-SA hta2-SA* genotype (also known as “*htaAQ*”) has previously been shown to increase both CPT and MMS sensitivity in *brc1*Δ cells^15^. This is consistent with Brc1 having known γ-H2A independent interactions at stalled forks. Previous genetic analysis has shown that Brc1 can bind to post-replication repair factor Rad18, which itself is recruited to stalled replication forks via an interaction with RPA coated ssDNA^16^. It has been suggested that this may aid stabilisation of RPA on its ssDNA substrate to inhibit HR-mediated fork resolution.

It has been shown that Brc1 can be prevented from binding to γ-H2A by the point mutation *brc1-T672A* and that this mutation prevents the formation of Nse4-GFP foci in response to MMS^13, 17^. We have since introduced the *brc1-T672A* allele into our Nse4-mEos3 strain and show a similar reduction in Nse4-mEos3 bound fraction compared to both (Supplementary Figure 7) *brc1*Δ and *hta1-SA hta2-SA* cells. This further strengthens our model that Brc1-dependent recruitment of Smc5/6 is limited to regions of γ-H2A, but this is not the only role of Brc1 at stalled forks.

7. Figure 5, the Alt 2017 paper suggests that while smc5-R609, 615E abolishes ssDNA binding, smc5-Y612G or smc6-x, and to a large extent Smc6-F528A, exhibit close to wild-type level of ssDNA binding. Here, these mutants showed similar Nse4 sptPALM behavior. Thus, these observations would argue that features other than ssDNA binding affect Nse4-chromatin association.

This reviewer is correct when stating that Alt et al. (2017) show that *smc5R609E R615E* severely abrogates initial ssDNA binding in vitro. Whilst the hinge destabilisation (*smc5-Y612G*) and secondary ssDNA interaction (*smc6-F528A*) mutants have more mild effects on ssDNA binding, they affect the kinetics of association and dissociation with ssDNA.

Importantly, they are also display significant sensitivity to DNA damaging agents and replication stress. However, unlike the dsDNA binding and ATPase mutants our SPT analysis demonstrates that this does not alter the bound fraction of Smc5/6 in unchallenged cells. This is one of the central points of our manuscript.

We have taken steps to highlight the different in vitro binding characteristics of these mutants in the manuscript to make it clearer for the reader. Furthermore, we have performed SPT analysis on the *smc5R609E R615E* mutant in the presence of MMS (Supplementary Figure 5). Interestingly, we found that the modest increase in bound fraction seen in response to MMS in wild type cells is either reduced or prevented in these hinge mutants.

8. Figure 6B: The fluctuation tests lack error bars. Since Figure S6 is labeled as a repeat of Figure 6B left, I assume these are single determinations. The data should be reported as mean of n=3. The rate difference between Figure 6B left and Figure S6 are unusually large for a rate determination, which is usually quite robust. All the more a reason to bolster these data with n=3.

We have repeated these experiments so that there are now three repeats for each strain. We have also performed this analysis on a further mutant (*nse3-R254E*) to strengthen our observation that mutants that are defective in chromatin association produce distinct outcomes at HR-dependent processes than ssDNA interaction mutants.

Text changes or clarifications:9. Column graphs should be converted to scatter "superplots" as recently proposed (Lord et al., 2020) to show individual cell data and variability between experiments.

In our study, we performed 3 technical repeats (fields of view of the same sample) for each biological repeat. In each technical repeat the single particle trajectories recorded are pooled together to populate the Spot-On displacement histograms. It is a standard process in the field to combine trajectories from individual nuclei from smaller eukaryotic cells as it would not possible to perform “superplots” of each nucleus as there would not be enough data to create a histogram and robustly perform the kinetic modelling. This exact approach has been recently applied to transcription factors in budding yeast cells and published in *eLife*^4^. However, we agree that it is important to highlight the variability between technical repeats and have thus altered our column graphs to display the fraction bound value extracted from each.

10. Line 35: insert 'function of the' in front of Smc5/6.

This has been altered.

11. Line 42: It might help the reader to include the statement here, which is found in the beginning of the discussion (lines 355 ff), to indicate that SMC must have functions beyond HR.

This is a good suggestion and we have included a statement in the new text.

12. Figure 3A: The evidence that Brc1 is binding to gammaH2AX should be discussed.

This has now been highlighted and is now figure 5A.

[Editors’ note: what follows is the authors’ response to the second round of review.]

Essential revisions:The authors report in their rebuttal that they have performed SPT at hourly intervals in wild type cells and found that the fraction bound was highest at 4-5 hours. These data should be included in the manuscript, maybe as a supplemental figure.

We have incorporated the data for our 5-hour MMS treatment as a supplementary figure (S9) and have included the relevant figure legend in the text.

References:

1. Hallett, S. T. et al. Nse5 / 6 is a negative regulator of the ATPase activity of the Smc5 / 6 complex. (2021).

2. Steigenberger, B., Schaefer, I., Soh, Y., Scheltema, R. A. and Gruber,S. Nse5 / 6 inhibits the Smc5 / 6 ATPase to facilitate DNA substrate selection. (2021).

3. ansen, A. S. et al. Robust model-based analysis of single-particle tracking experiments with spot-on. *eLife* 7, 1–33 (2018).

4. Ranjan, A. et al. Live-cell single particle imaging reveals the role of RNA polymerase II in histone H2A.Z eviction. *eLife* 9, 1–15 (2020).

5. Crabben, S. N. Van Der et al. Destabilized SMC5 / 6 complex leads to chromosome breakage syndrome with severe lung disease Find the latest version: Destabilized SMC5 / 6 complex leads to chromosome breakage syndrome with severe lung disease. 1–12 (2016) doi:10.1172/JCI82890.Recently.

6. Alt, A. et al. Specialized interfaces of Smc5/6 control hinge stability and DNA association. Nat. Commun. 8, 14011 (2017).

7. Zabrady, K. et al. Chromatin association of the SMC5/6 complex is dependent on binding of its NSE3 subunit to DNA. Nucleic Acids Res. 44, 1064–1079 (2016).

8. Marguerat, S. et al. Quantitative analysis of fission yeast transcriptomes and proteomes in proliferating and quiescent cells. Cell (2012) doi:10.1016/j.cell.2012.09.019.

9. Elbatsh, A. M. O. et al. Cohesin Releases DNA through

Asymmetric ATPase-Driven Ring Opening. Mol. Cell 61, 575–588 (2016).

10. Hassler, M. et al. Structural Basis of an Asymmetric Condensin ATPase Cycle. Mol. Cell 74, 1175-1188.e9 (2019).

11. Verkade, H. M., Bugg, S. J., Lindsay, H. D., Carr, A. M. and O’Connell, M. J. Rad18 is required for DNA repair and checkpoint responses in fission yeast. Mol. Biol. Cell 10, 2905–2918 (1999).

12. Lee, K. M. et al. Brc1-mediated rescue of Smc5/6 deficiency: Requirement for multiple nucleases and a novel Rad18 function. Genetics (2007) doi:10.1534/genetics.106.067801.

13. Oravcová, M. et al. Brc1 Promotes the Focal Accumulation and SUMO Ligase Activity of Smc5-Smc6 during Replication Stress. Mol. Cell.

Biol. 39, 1–15 (2018).

14. Wan, B., Wu, J., Meng, X., Lei, M. and Zhao, X. Molecular Basis for

Control of Diverse Genome Stability Factors by the Multi-BRCT Scaffold Rtt107. Mol. Cell (2019) doi:10.1016/j.molcel.2019.05.035.

15. Mejia-Ramirez, E., Limbo, O., Langerak, P. and Russell, P. CriticalFunction of H2A in S-Phase. PLoS Genet.(2015) doi:10.1371/journal.pgen.1005517.

16. Reubens, M. C., Rozenzhak, S. and Russell, P. Multi-BRCT Domain

Protein Brc1 Links Rhp18/Rad18 and H2A To Maintain Genome Stability during S Phase. Mol. Cell. Biol. (2017) doi:10.1128/mcb.0026017.

17. Williams, J. S. et al. h2A binds Brc1 to maintain genome integrity during S-phase. EMBO J. (2010) doi:10.1038/emboj.2009.413.